# ALMEA: ACTIVE LEARNING FOR MULTIMODAL ENTITY ALIGNMENT WITH SEMANTIC IMPUTATION

## ABSTRACT

Multimodal knowledge graphs (MMKGs) offer enriched knowledge representation by integrating structural, visual and textual information from heterogeneous sources. However, existing multimodal entity alignment (MMEA) approaches face significant challenges due to missing modalities and semantic inconsistencies across sources. These limitations compromise alignment robustness, especially in low-resource scenarios with limited seed pairs (i.e. manually annotated aligned entities as supervision). To bridge the gap, we propose **Active Learning for Multimodal Entity Alignment with Semantic Imputation (ALMEA)**, a MMEA framework that integrates semantic calibration and active learning to improve alignment. Specifically, ALMEA synthesizes embeddings for missing modalities and refines semantic representations to address inconsistencies across MMKGs. This approach iteratively selects optimal candidate pairs within the learnable budget through active learning strategies, thereby acquiring richer modal information in low-resource scenarios. On the benchmark MMKG dataset, experimental results indicate that ALMEA consistently outperforms state-of-the-art baseline models under the low-resource scenario, achieving average improvements of 5.16% in Mean Reciprocal Rank (MRR) and 5.57% in Hits at Top-1 (Hits@1). Our anonymized code is available at github.com/RTX4090123/ALMEA.

## 1 INTRODUCTION

As multimodal knowledge graphs (MMKGs) become more prevalent, multimodal entity alignment (MMEA) becomes critical for integrating knowledge across heterogeneous sources. A range of MMEA approaches focus on fusing relational, attribute, and visual modalities to ensure consistent entity alignment across heterogeneous sources. For instance, Chen et al. Chen et al. (2020a) proposed fusing these modalities into a unified embedding space, while EVA Liu et al. (2021b) prioritized visual similarity for alignment. MSNEA Chen et al. (2022) enhanced cross-modal interaction with visual cues, and MCLEA Lin et al. (2022) applied contrastive learning to refine intra-modal representations. However, these methods tend to overly rely on visual data, limiting their effectiveness when visual modalities are absent. To improve cross-modal semantic complementarity, the MEAformer Chen et al. (2023a) incorporated a multi-head attention mechanism Vaswani (2017) to dynamically adjust modality weights, minimize information loss, and extract core semantics for alignment. More recently, GEEA Guo et al. (2023) introduced a VAE-based generative framework to reconstruct embeddings for missing modalities between MMKGs, improving alignment by strengthening feature associations and facilitating information sharing.

Despite advancements in missing modality handling Chen et al. (2023b), maintaining **semantic consistency** across heterogeneous MMKGs remains challenging, particularly when imputing missing modalities. As shown in Figure 1(a), MEAformer misallocates modality weights when key inputs (e.g., visual data for "Zachary Quinto") are absent, over-relying on noisy attributes or neighbors. Structural disparities (e.g., 190 vs. 8 neighbors) further magnify misalignment, revealing MEAformer's limitations under cross-KG heterogeneity. Moreover, limited seed pairs due to annotation constraints create a critical **low-resource challenge** in MMEA, resulting in poor alignment robustness Ni et al. (2023). As Figure 1(b) shows, existing methods falter under sparse supervision, e.g., "Mayor" in FB15K ranks only 30th with 5% seeds, reflecting poor semantic relevance. Conversely, ALMEA achieves correct alignment with just 2% budget by leveraging semantic cues from related domains

| Source Entity | MEAformer (w/o Active Learning 5%) | ALMEA (w Active Learning 5% + 2 %) |
|---|---|---|
| Mayor | Rank 1: Europe
Rank 2: University
Rank 3: Statistics
...
**Rank 30: Mayor** | **Rank 1: Mayor**
Rank 2: French and ...
Rank 3: Civil Engineer
... |
| Celebrity Fit Club | Rank 1: The Ultimate Fighter TV
Rank 2: America's Next Top Model
Rank 3: Charmed
...
**Rank 8: Celebrity Fit Club** | **Rank 1: Celebrity Fit Club**
Rank 2: Santa Barbara (TV series)
Rank 3: The_Waltons
... |
| Leiden University | Rank 1: Delft University of Technology
Rank 2: University of Kiel
Rank 3: Victoria University of Wellington
...
**Rank 5: Leiden University** | **Rank 1: Leiden University**
Rank 2: University of Kiel
Rank 3: Bates College
... |

(a)                              (b)

Figure 1: Challenges in MMEA: (a) semantic inconsistencies from missing modalities, notably absent visual data in FB15K and DB15K  García-Durán & Niepert (2018); (b) limited training pairs, reducing alignment accuracy and robustness. Case study on FB15K-DB15K with 5% seeds shows ALMEA, with 2% actively selected pairs, outperforms MEAformer in alignment quality. For more case studies, see Appendix D.

(e.g., politics, professions). For entities such as "Celebrity Fit Club" and "Leiden University", both models find relevant candidates, but ALMEA's active selection injects richer semantics.

To address these challenges, we propose **ALMEA**, a framework addressing low-resource challenges by combining semantically calibrated missing modality imputation with active learning. ALMEA comprises three core modules: (i) *Latent Semantic Learning (LSL)* synthesizes embeddings for missing modalities by leveraging available ones, preserving the distributional semantics of entities within the MMKG. (ii) Latent Semantic Calibration (LSC) aligns semantically inconsistent entities by adjusting modality contributions for robust cross-KG alignment despite missing modalities. (iii) Active Candidate Selection (ACS) iteratively identifies high-quality, representative, and diverse entity pairs via optimization to mitigate low-resource limitations. Extensive experiments demonstrate the robustness and effectiveness of ALMEA, outperforming state-of-the-art MMEA baselines by up to 5.16% in MRR and 5.57% in Hits@1 under a low-resource scenario. Under other scenarios, it still achieves 3.58% and 3.42% gains. Ablation studies highlight its capability to bridge semantic gaps and handle incomplete MMKGs to yield improvements in robustness by 2.02% and 2.85% over non-active variants.

Our main contributions are as follows:

- We propose **ALMEA**, an MMEA framework that calibrates missing modalities and leverages active learning to improve alignment accuracy and robustness.

- To tackle semantic inconsistencies, ALMEA calibrates and aligns the precise semantics of entities with missing modalities across MMKGs.

- The efficient active learning component improves low-resource learning by iteratively selecting optimal candidate pairs within a learnable budget.

- Experiments demonstrate that ALMEA outperforms existing baseline models in robustness, optimistic speed and validate the effectiveness of active learning in low-resource scenarios.

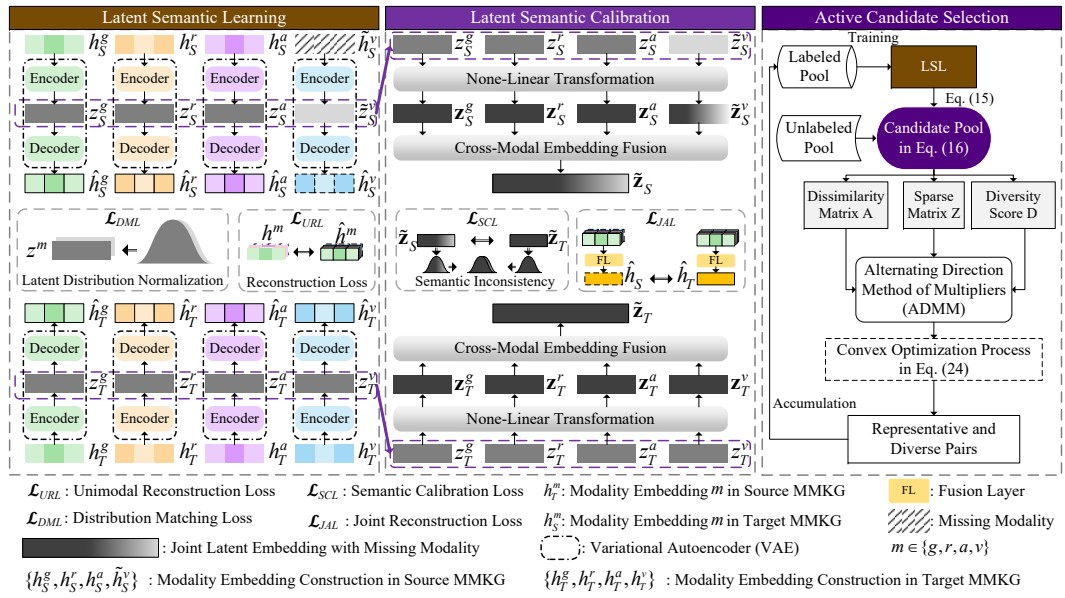

Figure 2: ALMEA Architecture. (i) Latent Semantic Learning (LSL) preserves distributional semantics via compressed cross-modal embeddings within MMKGs; (ii) Latent Semantic Calibration (LSC) calibrates semantic consistency across MMKGs; (iii) Active Candidate Selection (ACS) selects optimal candidate pairs to enhance representation robustness.

## 2 PRELIMINARIES

**Multimodal Knowledge Graph (MMKG).** A multimodal knowledge graph is represented as $\mathcal{G} = \{\mathcal{E}, \mathcal{R}, \mathcal{A}, \mathcal{V}, \mathcal{T}\}$, where $\mathcal{E}$ is the set of entities, $\mathcal{R}$ denotes relationships between entities, $\mathcal{A}$ represents attributes, and $\mathcal{V}$ represents the visual information of entities. $\mathcal{T}$ is a set of relational triples $(e_h, r, e_t) \in \mathcal{T}$, with $e_h, e_t \in \mathcal{E}$ and $r \in \mathcal{R}$.

**Multimodal Entity Alignment (MMEA).** Consider two MMKGs: the source $\mathcal{G}_S = \{\mathcal{E}_S, \mathcal{R}_S, \mathcal{A}_S, \mathcal{V}_S, \mathcal{T}_S\}$ and the target $\mathcal{G}_T = \{\mathcal{E}_T, \mathcal{R}_T, \mathcal{A}_T, \mathcal{V}_T, \mathcal{T}_T\}$. Let $L$ denote a small set of known equivalent entity pairs, called seed pairs, and $U$ represent the set of entities with unknown counterparts. Given $\mathcal{G}_S$, $\mathcal{G}_T$, and $L$, the goal of the MMEA task is to identify and align equivalent entity pairs between $\mathcal{G}_S$ and $\mathcal{G}_T$ for $U$, i.e., $\{(e_i, e_j) \mid e_i \in \mathcal{E}_S^U, e_j \in \mathcal{E}_T^U, e_i \equiv e_j\}$, where $\equiv$ denotes entity equivalence across MMKGs.

**Multimodal Knowledge Encoder.** Each entity $e_i$ is typically associated with pre-trained modality embeddings $\{x_i^g, x_i^r, x_i^a, x_i^v\}$ for graph, relational, attribute, and visual modalities. Pretrained encoders $\mathcal{M}^m$, which takes $x_i^m$ as inputs and obtain modality embeddings $h_i^m = \mathcal{M}^m(x_i^m)$ for each modality $m \in \{g, r, a, v\}$ Lin et al. (2022). We follow the prior MMEA works (e.g., Chen et al. (2023a), Li et al. (2024)), where the cross-modal embedding of entity $e_i$ is obtained by projecting modality features to a shared space and fusing them via normalized and weighted aggregation $h_i = \sum_{m \in \{g,r,a,v\}} \pi_m \cdot \text{Norm}(h_i^m)$, $\pi_m = \frac{\exp(w_m)}{\sum_j \exp(w_j)}$, where the modality weight $\pi_m$ is derived from the learnable parameter $w_m$.

## 3 METHODOLOGY

We propose ALMEA, a MMEA framework integrating semantic calibration and active learning to enhance alignment. Figure 2 outlines its three core components, detailed in subsequent sections.

## 3.1 LATENT SEMANTIC LEARNING (LSL)

Missing modalities challenges the preservation of robust latent semantics in synthesized embeddings. To address this, we employ a Variational Autoencoder (VAE) to reconstructs missing modalities via latent distribution learning, where imputation is simulated through masking and Gaussian sampling.

**Latent Distributional Semantics.** To identify missing modalities, we introduce a binary masking matrix $\mathbf{B} \in \{0, 1\}^{d \times M}$, where $M$ denotes the number of modalities. Each column vector $b^m \in \{0, 1\}$ represents modality $m$ and indicates element-wise preservation (1) or masking (0) as follows:

$$\tilde{h}^m = b^m \cdot h^m, \quad b^m = \begin{cases} 1 & \text{if modality } m \text{ is preserved} \\ 0 & \text{if modality } m \text{ is masked} \end{cases}, \quad m \in \{g, r, a, v\}. \tag{1}$$

Next, the Encoder takes masked modality embedding $h^m$, thereby producing hidden representations $\mathbf{h}^m = \text{Encoder}(\tilde{h}^m)$ for each modality $m \in \{g, r, a, v\}$. These representations are then transformed into latent variables $z^m$ by computing the mean $\mu^m = \text{Linear}_\mu(\mathbf{h}^m)$ and standard deviation $\sigma^m = \text{Linear}_\sigma(\mathbf{h}^m)$ via fully connected layers. Using the reparameterization trick, $z^m$ is sampled with added Gaussian noise $\epsilon \in \mathcal{N}(\mathbf{0}, \mathbf{I})$ for backpropagation: $\tilde{z}^m = \text{Linear}_\mu(\tilde{\mathbf{h}}^m) + \text{Linear}_\sigma(\tilde{\mathbf{h}}^m) \odot \epsilon$ where $\tilde{z}^m$ is the latent embedding sampled from a Gaussian distribution for modality $m \in \{g, r, a, v\}$.

## 3.2 LATENT SEMANTIC CALIBRATION (LSC)

To address semantic inconsistencies in the synthesized modality, we introduce the Latent Semantic Calibration (LSC) layer, which refines the reconstructed semantic details for missing modalities across MMKGs in the compressed embedding space. Following latent semantic normalization in the LSL layer (Section 3), LSC aligns the latent distributions of entities with missing modalities to their counterparts, reducing inconsistencies across MMKGs.

**Latent Embedding Extraction.** Following latent distribution alignment, masked and unmasked modality embeddings $\tilde{z}^m$ and unmasked $z^m$ modalities are derived via the reparameterization trick. To calibrate latent semantics across modalities, we employ a shared linear transformation followed by Tanh activation for missing modalities $m$ as $\{\mathbf{z}^m, \tilde{\mathbf{z}}^m\} = \tanh(W_m\{z^m, \tilde{z}^m\} + b_m)$ where $\tilde{\mathbf{z}}^m$ and $\mathbf{z}^m$ denote the calibrated latent embeddings of masked and unmasked modalities.

**Cross-Modal Embedding Fusion.** To capture modality-specific semantic contributions, we compute weights as: $\tilde{\mathbf{w}}^m = \text{MLP}(\tilde{\mathbf{z}}^m) / \sum_{m \in \{g,r,a,v\}} \text{MLP}(\mathbf{z}^m)$, where $\tilde{\mathbf{w}}^m$ denotes the weight vector for the calibrated latent embedding of masked modality $m$ (e.g., $\tilde{\mathbf{w}}^v$). The joint latent embedding are then fused as $\tilde{\mathbf{z}} = \sum_{m \in \{g,r,a,v\}} \mathbf{z}^m \odot \mathbf{w}^m$ where $\tilde{\mathbf{z}}^m$ denotes the calibrated latent embedding of the masked visual modality $\tilde{\mathbf{z}}^v$, while $\mathbf{z}^m$ represents calibrated embeddings from unmasked modalities. where $\tilde{\mathbf{z}}^m$ denotes the calibrated latent embedding of the masked visual modality $\tilde{\mathbf{z}}^v$, while $\mathbf{z}^m$ for $m \in \{g, r, a, v\}$ represents calibrated embeddings from unmasked modalities.

**Cross-Modal Calibration Loss.** To bridge the semantic gap across MMKGs, we compare the masked joint latent embedding $\tilde{\mathbf{z}}_S \in \mathcal{E}_S$ from the source MMKG and the unmasked joint latent embedding $\mathbf{z}_T \in \mathcal{E}_T$ from the target MMKG. Both embeddings are normalized via ReLU activation followed by L1 normalization, yielding effective semantic probability distributions. This comparison minimizes semantic discrepancies using the Kullback-Leibler (KL) divergence:

$$\mathcal{L}_{SCL}(\tilde{\mathbf{z}}_S, \mathbf{z}_T) = D_{KL}(\tilde{\mathbf{z}}_S \| \mathbf{z}_T) + D_{KL}(\mathbf{z}_T \| \tilde{\mathbf{z}}_S) \tag{2}$$

where $\mathcal{L}_{SCL}$ enforces consistency between source and target semantic distributions. $\tilde{\mathbf{z}}_S(x)$ and $\mathbf{z}_T(x)$ denote the probabilities of the $x$-th dimension in the source and target distributions, respectively.

**Joint Learning Objective.** The learning objective ensures effective semantic calibration and accurate reconstruction of missing modality semantics. To clearly illustrate the role of each loss component, we summarize the overall optimization as a meta-algorithm in Algorithm 1.

## 3.3 ACTIVE CANDIDATE SELECTION (ACS)

To mitigate low-resource supervision, we leverage an active learning component (ACS). ACS adaptively selects a small, representative, diverse set of candidate entity pairs under a labeling budget.

---

**Algorithm 1** Meta Loss Design for Model $M$

---

1: **Input:** modality $m \in \{g, r, a, v\}$; original embedding $h^m$; calibrated latent embeddings $\tilde{\mathbf{z}}^m$ (masked), $\mathbf{z}^m$ (unmasked); Gaussian parameters $(\mu^m, \sigma^m)$.
2: **Step 1 (core): Cross-Modal Calibration Loss** $\rightarrow \mathcal{L}_{\text{SCL}}$ (Eq. 2)
3: **Step 2: Latent Distribution Normalization** $\rightarrow \mathcal{L}_{DML}$ (Appendix B.2, Eq. 9)
4: **Step 3: Missing Modality Reconstruction** $\rightarrow \mathcal{L}_{URL}$ (Appendix B.2, Eq. 10)
5: **Step 4: Cross-Modal Alignment** $\rightarrow \mathcal{L}_{JAL}$ (Appendix B.2, Eq. 11)
6: **Final Objective** $\rightarrow \mathcal{L} = \lambda_1 \mathcal{L}_{\text{SCCL}} + \lambda_2 \mathcal{L}_{\text{DML}} + \lambda_3 \mathcal{L}_{\text{URL}} + \lambda_4 \mathcal{L}_{\text{JAL}}$
7: **Output:** trained model $M$.

---

This strategy improves alignment robustness while effectively controlling labeling cost. Specifically, we adapt the Dissimilarity-based Sparse Subset Selection (DS3) objective Elhamifar et al. (2015), renowned for its effectiveness in representative and diverse sample selection, as our acquisition rule. ACS iteratively promotes selected pairs from the unlabeled pool to the labeled pool to refine the training process.

**Candidate Pool $Q_{sorted}$.** After obtaining semantically calibrated embeddings for all entities, the active learning module iteratively refines embedding quality. Let $\mathcal{E}_S = \{h_{S_1}, h_{S_2}, \ldots, h_{S_M}\}$ and $\mathcal{E}_T = \{h_{T_1}, h_{T_2}, \ldots, h_{T_N}\}$ denote the joint embeddings of source and target entities. To identify reliable candidate pairs, we compute angular distances between entities in $\mathcal{E}_S$ and $\mathcal{E}_T$. Since nearest-neighbor (NN) relations are asymmetric, we construct a symmetric candidate set via mutual NN:

$$Q = \big\{ (S, T) : \arg\min_T \|h_S - h_T\|^2 = \arg\min_S \|h_T - h_S\|^2 \big\} \tag{3}$$

where $h_S \in \mathcal{E}_S, h_T \in \mathcal{E}_T$. We rank candidates by similarity to obtain $Q_{\text{sorted}} = \text{sort}(Q)$.

**The Representativeness Perspective.** Let $Z \in \mathbb{R}^{K \times K}$ be an indicator matrix where $\mathbf{z}_{(q_i)(q_j)}$ denotes the probability that candidate pair $q_j$ is represented by $q_i$, with $\{q_i, q_j \in Q_{\text{sorted}}\}$. Selecting a small representative set corresponds to inducing row sparsity in $Z$, which we promote via the mixed norm defined as follows:

$$\|Z\|_{2,1} = \sum_{i=1}^{d} \|\mathbf{z}_{i,:}\|_2 \tag{4}$$

where $\mathbf{z}_i$ denotes the $i$ row of $Z$, and $\mathbf{z}_i \neq 0$ implies that $q_i$ serves as a representative for at least one $q_j$. Larger $\mathbf{z}_{(q_i)(q_j)} \in [0, 1]$ indicates higher assignment probability of $q_j$ to $q_i$.

To quantify dissimilarity between candidate pairs in the same embedding space, we define the dissimilarity matrix $A$ as follows:

$$A[i, j] = \tfrac{1}{2} \big( \|h_{S_i} - h_{S_j}\|^2 + \|h_{T_i} - h_{T_j}\|^2 \big). \tag{5}$$

where $(S_i, T_i)$ and $(S_j, T_j)$ are candidate pairs from $Q_{\text{sorted}}$. A smaller value of $A[i, j]$ indicates a higher likelihood that $q_i$ represents $q_j$.

**The Diversity Perspective.** After sparsifying the indicator matrix $Z$ via the dissimilarity matrix $A$, the selection process favors a smaller set of representative candidates from $Q_{\text{sorted}}$. To ensure broader information coverage in the labeled set $L$, we introduce the diversity matrix $C$, defined as follows:

$$C[i, j] = \tfrac{1}{2} \big( \|h_{S_i} - h_{L_j^S}\|^2 + \|h_{T_i} - h_{L_j^T}\|^2 \big). \tag{6}$$

where $(S_i, T_i)$ is the candidate pair from $Q_{\text{sorted}}$, while $(L_j^S, L_j^T)$ is labeled pair from $L$.

To balance diversity and representativeness, we incorporate the diversity score $D_{q_i} \in [1, \sigma]$ into the optimization, defined as follows:

$$D_{q_i} = \sigma - (\sigma - 1) \times \big( \min_{l \in L} C_{li} \big/ \max_{q \in Q} \min_{l \in L} C_{lq} \big) \tag{7}$$

Ideally, higher diversity scores indicates greater dissimilarity from labeled pairs, thus increasing its likelihood of selection.

**Optimization.** Incorporating diversity, the optimization problem is reformulated as:

$$\min_Z \lambda \|DZ\|_{2,1} + \text{tr}(A^T Z) \quad \text{s.t.} \quad \mathbf{1}^T Z = \mathbf{1}^T, \quad Z \geq 0 \tag{8}$$

where the diversity score $D$ acts as a penalty term to encourage the selection of both representative and diverse unlabeled samples. Derivations and a sensitivity study of $\lambda$ appear in Appendix B.1-B.4,C.3. Algorithm 2 summarizes the overall pipeline in Appendix B.5.

Table 1: Alignment Accuracy Comparison. Results are under 20%, 50%, and 80% valid alignment pairs settings. The best results are bolded, the second best are underlined, and ('*') indicates reproduction via code implementation. The ↑ indicates performance improvement over SOTA.

| Methods | FB15K-DB15K (20%) | | | FB15K-DB15K (50%) | | | FB15K-DB15K (80%) | | |
|---|---|---|---|---|---|---|---|---|---|
| | MRR(%) | Hits@1(%) | Hits@10(%) | MRR(%) | Hits@1(%) | Hits@10(%) | MRR(%) | Hits@1(%) | Hits@10(%) |
| TransE | 13.4 | 7.8 | 24.0 | 30.6 | 23.0 | 44.6 | 50.7 | 42.6 | 65.9 |
| GCN-align | 8.7 | 5.3 | 17.4 | 29.3 | 22.6 | 43.5 | 47.2 | 41.4 | 63.5 |
| SEA | 25.5 | 17.0 | 42.5 | 47.0 | 37.3 | 65.7 | 50.5 | 51.2 | 78.4 |
| MMEA | 35.7 | 26.5 | 54.1 | 51.2 | 41.7 | 70.3 | 68.5 | 59.0 | 86.9 |
| EVA* | 18.48 ± 0.39 | 11.90 ± 0.27 | 31.60 ± 0.51 | 26.14 ± 0.47 | 18.06 ± 0.47 | 42.32 ± 0.59 | 35.34 ± 0.51 | 25.98 ± 0.46 | 53.48 ± 0.44 |
| MSNEA* | 37.89 ± 0.35 | 28.74 ± 0.26 | 55.78 ± 0.34 | 62.85 ± 6.28 | 54.76 ± 7.48 | 75.56 ± 6.21 | 78.97 ± 5.90 | 73.44 ± 6.65 | 88.53 ± 4.08 |
| MCLEA* | 40.80 ± 0.15 | 31.18 ± 0.14 | 59.28 ± 0.13 | 68.27 ± 0.32 | 60.50 ± 0.55 | 81.99 ± 0.71 | 80.99 ± 0.09 | 75.39 ± 0.26 | 90.11 ± 0.22 |
| GEEA* | 40.23 ± 0.12 | 30.12 ± 0.18 | 60.34 ± 0.22 | 64.84 ± 0.18 | 55.99 ± 0.14 | 81.01 ± 0.33 | 79.46 ± 0.74 | 73.51 ± 0.006 | 89.45 ± 0.66 |
| MEAformer* | 51.84 ± 0.15 | 41.69 ± 0.19 | 71.35 ± 0.13 | 69.79 ± 0.02 | 61.70 ± 0.07 | 84.02 ± 0.09 | 81.97 ± 0.09 | 76.39 ± 0.13 | 91.22 ± 0.08 |
| OTMEA* | 44.68 ± 5.06 | 37.47 ± 4.40 | 58.49 ± 6.36 | 65.00 ± 3.31 | 58.57 ± 3.25 | 76.21 ± 3.39 | 77.59 ± 3.79 | 72.41 ± 4.54 | 86.43 ± 2.28 |
| SimDiff* | 54.31 ± 0.25 | 44.39 ± 0.27 | 72.99 ± 0.19 | 71.56 ± 0.02 | 63.58 ± 0.05 | 85.42 ± 0.01 | 83.61 ± 0.08 | 78.74 ± 0.03 | 92.03 ± 0.03 |
| ALMEA w/o ACS | **56.00** ± 0.12 | **46.81** ± 0.14 | **73.52** ± 0.32 | **74.20** ± 0.27 | **67.41** ± 0.20 | **86.31** ± 0.45 | **84.83** ± 0.03 | **80.45** ± 0.21 | **93.28** ± 0.18 |
| Improvement | ↑ 1.69 | ↑ 2.42 | ↑ 0.53 | ↑ 2.64 | ↑ 3.83 | ↑ 0.89 | ↑ 1.22 | ↑ 1.61 | ↑ 1.25 |
| ALMEA | **59.43** ± 0.10 | **50.79** ± 0.07 | **75.64** ± 0.27 | **74.16** ± 0.29 | **67.40** ± 0.32 | **86.16** ± 0.29 | **83.80** ± 0.16 | **79.08** ± 0.22 | **92.41** ± 0.48 |
| Improvement | ↑ 5.12 | ↑ 6.40 | ↑ 2.65 | ↑ 2.60 | ↑ 3.82 | ↑ 0.74 | ↑ 0.19 | ↑ 0.34 | ↑ 0.38 |

| Methods | FB15K-YAGO15K (20%) | | | FB15K-YAGO15K (50%) | | | FB15K-YAGO15K (80%) | | |
|---|---|---|---|---|---|---|---|---|---|
| | MRR(%) | Hits@1(%) | Hits@10(%) | MRR(%) | Hits@1(%) | Hits@10(%) | MRR(%) | Hits@1(%) | Hits@10(%) |
| TransE | 11.2 | 6.4 | 20.3 | 26.2 | 19.7 | 38.2 | 46.3 | 39.2 | 59.5 |
| GCN-align | 15.3 | 8.1 | 23.5 | 29.4 | 23.5 | 42.4 | 47.7 | 40.6 | 64.3 |
| SEA | 21.8 | 14.1 | 37.1 | 38.8 | 29.4 | 57.7 | 60.5 | 51.4 | 77.3 |
| MMEA | 31.7 | 23.4 | 48.0 | 48.6 | 40.3 | 64.5 | 68.2 | 59.8 | 83.9 |
| EVA* | 14.86 ± 0.24 | 9.38 ± 0.23 | 25.52 ± 0.31 | 23.36 ± 0.37 | 15.98 ± 0.31 | 37.76 ± 0.49 | 38.86 ± 0.34 | 30.12 ± 0.43 | 55.16 ± 0.57 |
| MSNEA* | 36.74 ± 2.24 | 28.25 ± 2.16 | 53.29 ± 2.41 | 58.30 ± 4.31 | 50.17 ± 4.43 | 73.93 ± 4.21 | 69.62 ± 6.69 | 62.57 ± 6.69 | 82.82 ± 6.32 |
| MCLEA* | 36.48 ± 0.12 | 28.35 ± 0.12 | 51.76 ± 0.19 | 59.36 ± 0.08 | 50.64 ± 0.18 | 74.40 ± 0.71 | 76.59 ± 0.16 | 70.61 ± 0.09 | 86.90 ± 0 .30 |
| GEEA* | 33.65 ± 0.06 | 25.38 ± 0.04 | 49.77 ± 0.01 | 57.59 ± 0.08 | 49.43 ± 0.005 | 72.65 ± 0.28 | 74.31 ± 0.08 | 67.59 ± 0.12 | 86.01 ± 0.13 |
| MEAformer* | 41.77 ± 0.02 | 32.73 ± 0.03 | 59.33 ± 0.07 | 63.74 ± 0.06 | 55.74 ± 0.10 | 78.02 ± 0.13 | 76.67 ± 0.04 | 70.46 ± 0.10 | 87.50 ± 0.16 |
| OTMEA* | 43.50 ± 0.95 | 35.21 ± 0.78 | 59.18 ± 1.37 | 62.97 ± 2.32 | 56.40 ± 2.08 | 74.62 ± 2.76 | 71.24 ± 8.74 | 65.12 ± 7.90 | 79.78 ± 8.69 |
| SimDiff* | 45.72 ± 0.04 | 36.61 ± 0.06 | 62.96 ± 0.05 | 64.35 ± 0.04 | 56.25 ± 0.06 | 78.69 ± 0.01 | 77.08 ± 0.03 | 71.30 ± 0.05 | 87.06 ± 0.03 |
| ALMEA w/o ACS | **46.09** ± 0.20 | **36.89** ± 0.24 | **63.81** ± 0.21 | **67.21** ± 0.09 | **59.61** ± 0.10 | **81.40** ± 0.35 | **80.10** ± 0.21 | **74.48** ± 0.16 | **90.16** ± 0.21 |
| Improvement | ↑ 0.37 | ↑ 0.28 | ↑ 0.85 | ↑ 2.86 | ↑ 3.36 | ↑ 2.71 | ↑ 3.02 | ↑ 3.18 | ↑ 3.10 |
| ALMEA | **50.92** ± 0.11 | **42.11** ± 0.16 | **67.70** ± 0.15 | **68.77** ± 0.10 | **61.33** ± 0.09 | **82.50** ± 0.21 | **78.34** ± 0.43 | **72.27** ± 0.53 | **88.95** ± 0.51 |
| Improvement | ↑ 5.20 | ↑ 5.50 | ↑ 4.74 | ↑ 4.42 | ↑ 5.08 | ↑ 3.81 | ↑ 1.26 | ↑ 0.97 | ↑ 1.89 |

# 4 EXPERIMENTS

We evaluate ALMEA's robustness and effectiveness, with additional results in Appendix CDE.

## 4.1 EXPERIMENTAL SETTINGS

**Datasets.** We evaluate the MMEA task on two benchmark datasets: FB15K-DB15K and FB15K-YAGO15K García-Durán & Niepert (2018), which include relational, visual, textual, and numerical attributes. The ground truth alignment pairs are split into 20%, 50%, and 80% for seed pairs. FB15K-DB15K has 94.94% of entities with visual modality information, while FB15K-YAGO15K has 81.29%.

**Baselines.** We compare our approach against two categories of baseline models: EA methods (TransE Bordes et al. (2013), GCN-Align Wang et al. (2018), and SEA Pei et al. (2019)) and MMEA methods (MMEA Chen et al. (2020a), EVA Liu et al. (2021b), MSNEA Chen et al. (2022), MCLEA Lin et al. (2022), GEEA Guo et al. (2023), MEAformer Chen et al. (2023a), OTMEA Wang et al. (2025) and SimDiff Li et al. (2024)).

**Evaluation Metrics.** We evaluate alignment performance using a pairwise distance metric. Hits@$n$ and Mean Reciprocal Rank (MRR) are used as evaluation measures.

**Implementation Detail.** We use a GAT-based encoder for structural features, GloVe-6B for textual features, and a VGG16 encoder for visual features, with all modality embeddings set to 300 dimensions. Additional hyperparameters and implementation details are in Appendix F.

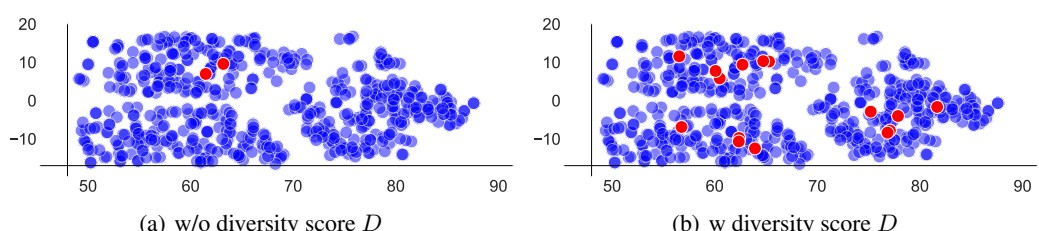

(a) w/o diversity score $D$         (b) w diversity score $D$

Figure 3: Effectiveness of Diversity in Active Learning. t-SNE visualizations show the sub-dataset (blue) and ACS-selected subset (red), comparing selections without and with the diversity score $D$.

## 4.2 OVERALL PERFORMANCE

To reduce bias, we report the mean and standard deviation over five independent runs for all models. Table 1 presents results for FB15K-YAGO15K (top) and FB15K-DB15K (bottom). "ALMEA w/o ACS" omits active learning, while "ALMEA" denotes the full model. Table 1 shows that ALMEA outperforms all baselines across training sizes (20%, 50%, and 80%) and evaluation metrics (MRR, Hits@1, Hits@10) on FB15K-DB15K and FB15K-YAGO15K. ALMEA without ACS improves MRR by 1.38%-2.89% (FB15K-DB15K) and 2.21%-4.07% (FB15K-YAGO15K), while the full model achieves larger gains of 1.83%-7.90% and 1.67%-9.15%, respectively, confirming robustness across training scales. These gains validate the efficacy of semantic calibration and active learning in entity alignment. ALMEA outperforms on FB15K-YAGO15K (4.19%) compared to FB15K-DB15K (1.25%), likely due to the lower availability of image modality in FB15K-YAGO15K. This reflects its robustness to missing image modalities and semantic inconsistencies, making it well-suited for incomplete MMKGs. The diminishing returns at higher training ratios (gains of 7.34%, 3.63%, and 1.90% for training sets of 20%, 50%, and 80%) highlights its particular advantage in low-resource scenarios through active candidate selection. For additional quantitative studies, see Appendix C.

## 4.3 ABLATION STUDY

LSL and LSC used 80% of the data to mitigate bias; ACS used 20% to assess low-resource efficacy.

## 4.4 IMPACT OF MODALITY REMOVAL.

Removing relational modality ('w/o relation modality') resulted in the most significant performance decline, decreasing by 4.02% on the FB15K-DB15K dataset and 4.11% on the FB15K-YAGO15K dataset. This is because relational information—particularly multi-to-one and multi-to-many relationships—is crucial for capturing complex dependencies between entities. By contrast, removing visual modality ('w/o visual modality') had the smallest impact, with performance declining by only 1.55% and 1.67% on the FB15K-YAGO15K dataset. This is primarily attributed to the weak cross-modal semantic contribution of visual modality (which operates solely on single entities) compared to other modalities, as relational modality information provides rich cross-modal information that compensates for the contribution of visual modality.

**Impact of Latent Semantic Learning (LSL).** As shown in Table 2, the "ALMEA w/o ACS (w/o LSL)" experiments result in significant MRR drop, 2.56% on FB15K-DB15K and 1.68% on FB15K-YAGO15K on average. This difference stems from the reliance on latent semantics, especially under limited modality availability in FB15K-YAGO15K.

**Impact of Latent Semantic Calibration (LSC).** The "ALMEA w/o ACS (w/o LSC)" results in Table 2 reveal that removing LSC degrades MRR by 3.51% (FB15K-DB15K) and 3.00% (FB15K-YAGO15K), suggesting that LSC is critical to refine the semantic details of the synthesized modalities. These findings highlight LSC's key role in mitigating semantic inconsistencies across MMKGs. Furthermore, both the semantic calibration loss (Formula (2)) and distribution match loss (Formula (9)) reaffirm LSC's indispensability. The Distribution Matching Loss constructs and smooths cross-modal latent representations; its absence weakens latent semantic alignment capabilities and impedes robust representation learning. Similarly, removing the semantic alignment loss would constrain

ALMEA's ability to extract latent semantics, thereby diminishing its effectiveness in capturing meaningful information during the MMEA task. This outcome aligns with our original design intent for the LSC component.

**Impact of Active Candidate Selection (ACS).** ALMEA w ACS experimental results demonstrate that removing the diversity score $D$ leads to a significant performance degradation: on the FB15K-DB15K and FB15K-YAGO15K datasets, performance decreased by 4.77% and 5.90% respectively. As illustrated in Figure 3(a) and (b), omitting $D$ (Figure 3(a)) alters the sparsity direction, resulting in an ACS-selected subset lacking diversity. This undermines performance, particularly in low-resource scenarios. Conversely, incorporating $D$ (Figure 3(b)) optimises sparsity direction while preserving diversity in the ACS-selected subset, effectively balancing representativeness and diversity. These results highlight the critical role of the diversity score $D$ in optimising active learning, particularly when training data is limited. For masking sensitivity analysis, see Appendix E.

Table 2: Ablation Study. The ↓ indicates performance degradation relative to the full model in bold.

| Variants | FB15K-DB15K (80%) | | | FB15K-YAGO15K (80%) | | |
|---|---|---|---|---|---|---|
| | MRR (%) | Hits@1 (%) | Hits@10 (%) | MRR (%) | Hits@1 (%) | Hits@10 (%) |
| **ALMEA w/o ACS** | **84.83** | **80.45** | **93.28** | **80.10** | **74.48** | **90.16** |
| w/o LSL | 82.80 (↓2.03) | 77.13 (↓3.32) | 90.94 (↓2.34) | 78.43 (↓1.67) | 72.45 (↓2.03) | 88.83 (↓1.33) |
| w/o LSC | 81.73 (↓3.10) | 76.41 (↓4.04) | 89.89 (↓3.39) | 77.06 (↓3.04) | 71.21 (↓3.27) | 87.48 (↓2.68) |
| w/o Distribution Match Loss | 82.96 (↓1.87) | 77.80 (↓2.65) | 91.36 (↓1.92) | 78.33 (↓1.77) | 72.30 (↓2.18) | 88.86 (↓1.3) |
| w/o Uni-modal Reconstruction Loss | 83.46 (↓1.37) | 78.50 (↓1.95) | 91.51 (↓1.77) | 78.56 (↓1.54) | 72.67 (↓1.81) | 88.99 (↓1.17) |
| w/o Semantic Calibration Loss | 82.13 (↓2.70) | 76.83 (↓3.62) | 90.24 (↓3.04) | 77.40 (↓2.70) | 71.53 (↓2.95) | 87.53 (↓2.63) |
| w/o Joint Alignment Loss | 83.43 (↓1.40) | 78.25 (↓2.20) | 91.38 (↓1.90) | 78.33 (↓1.77) | 72.45 (↓2.03) | 88.72 (↓1.44) |
| w/o Relation Modality | 80.93 (↓3.90) | 75.24 (↓5.21) | 90.34 (↓2.94) | 75.96 (↓4.14) | 68.61 (↓5.87) | 87.84 (↓2.32) |
| w/o Attribute Modality | 81.90 (↓2.93) | 76.24 (↓4.21) | 91.17 (↓2.11) | 76.66 (↓3.44) | 70.03 (↓4.45) | 88.00 (↓2.16) |
| w/o Vision Modality | 83.56 (↓1.27) | 78.58 (↓1.87) | 91.78 (↓1.50) | 78.46 (↓1.64) | 72.27 (↓2.21) | 89.01 (↓1.15) |
| Variants | FB15K-DB15K (20%) | | | FB15K-YAGO15K (20%) | | |
| | MRR (%) | Hits@1 (%) | Hits@10 (%) | MRR (%) | Hits@1 (%) | Hits@10 (%) |
| **ALMEA w ACS** | **59.43** | **50.79** | **75.64** | **50.92** | **42.11** | **67.70** |
| w/o Candidate Pool | 57.00 (↓2.43) | 48.07 (↓2.72) | 73.85 (↓1.79) | 47.16 (↓3.76) | 38.70 (↓3.41) | 63.67 (↓4.03) |
| w/o Diversity Score | 54.30 (↓5.13) | 44.70 (↓6.09) | 72.55 (↓3.09) | 44.86 (↓6.06) | 35.82 (↓6.29) | 62.35 (↓5.35) |

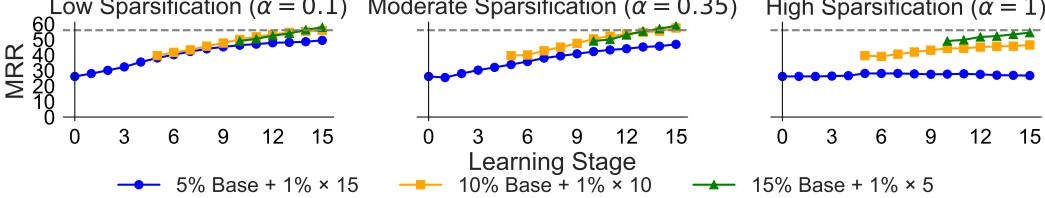

Figure 4: Active Learning under Varying Low-Resource Settings.

### 4.5 IMPACT OF VARYING LOW-RESOURCE SETTINGS

**Setup.** To evaluate the effectiveness of active learning, we conducted experiments under three low-resource settings: (**5% Base + 1% × 15**), (**10% Base + 1% × 10**), and (**15% Base + 1% × 5**), where each uses a small base (5-15%) of true alignment pairs and incrementally adds 1% candidate pairs per iteration. For example, (**5% Base + 1% × 15**) starts with 5% labeled seeds and performs 15 active learning rounds. To evaluate the impact of selected pair quality, we compared these against the empirical upper bound (**20% Base**), which randomly samples 20% of ground-truth pairs.

**Results.** Figure 4 shows FB15K-DB15K results under various low-resource settings. The setting (**15% Base + 1% × 5**) outperforms the empirical upper bound at sparsification factor $\alpha = 0.1$ and $\alpha = 0.35$, with improvements of 1.83% and 2.87%, respectively. This setting selects 15% of seed pairs randomly, with the remaining 5% strategically chosen via ACS. Despite limited initial base

information, ACS progressively enhances performance. However, higher active learning budgets, such as (**5% Base + 1% × 15**), yield diminishing returns. In contrast, (**15% Base + 1% × 5**) optimally balances base information and active learning, outperforming the empirical upper bound.

## 5 RELATED WORK

**Multimodal Entity Alignment.** Entity alignment across KGs is fundamental for graph fusion and downstream reasoning. Structural GNNs (e.g., GCN, GAT) aggregate local topology and semantics Kipf & Welling (2017); Veličković et al. (2018), while translation-based models (e.g., TransE, TransH) enforce triple-level geometric constraints Bordes et al. (2013); Wang et al. (2014). Multimodal EA (MMEA) fuses cross-modal representations to score entity similarity Chen et al. (2023a;b); Li et al. (2023b); Lin et al. (2022); Chen et al. (2022); Ni et al. (2023); Guo et al. (2023); Li et al. (2024); Wang et al. (2025). For instance, Chen et al. (2022) exploits visual semantics without GNNs, simplifying computation but weakening structural fidelity, whereas MEAformer Chen et al. (2023a) couples GAT/GNN encoders with Transformer attention, which can be brittle under missing modalities. SimDiffLi et al. (2024): It maps multiple modalities into a unified latent space and applies Gaussian noise with progressive denoising to enable intermodal information transfer and reduce heterogeneity. Open challenges persist in robustness to modality drop/noise, principled cross-modal fusion, cross-modal calibration, and sample efficiency.

**Missing Modality Imputation.** Several studies Li et al. (2023a); Guo et al. (2023); Chen et al. (2023b) have explored modality imputation. Li et al. Li et al. (2023a) used GNNs and generative algorithms for entity alignment. However, their method is sensitive to data quality and struggles with latent cross-modal semantics. Guo et al. Guo et al. (2023) applied VAE for modality reconstruction but struggled with source KG modality dependencies and limited semantic capture of missing modalities. Chen et al. Chen et al. (2023b) introduce an uncertainty-based method for approximating missing modalities, yet reconstruction accuracy is hindered by semantic ambiguity.

**Semantic Calibration.** Key studies addressing semantic distribution discrepancies for cross-modal alignment include UNITER Chen et al. (2020c), which uses a Transformer encoder for joint image-text embedding and masked semantic prediction. ALBEF Li et al. (2021), which employs contrastive learning to align semantically similar pairs; and ALIGN Cohen (1997), which maps features to probability distributions for distribution-level alignment. Building on these, we propose a latent semantic calibration module to bridge source-target semantic gaps and handle modality absence and inconsistency.

**Simulated Active Learning.** Recent work studies entity alignment in simulation-based active learning by selecting queries according to structural centrality, coverage, and uncertainty criteria Berrendorf et al. (2021); Liu et al. (2021a); Zeng et al. (2021); Huang et al. (2023). For instance, RAC Zeng et al. (2021) uses an RL policy over degree, PageRank, and information density; Liu et al. Liu et al. (2021a) maximize information gain via uncertainty scoring. Huang et al. (2023) refines candidate selection with similarity-margin constraints. However, these methods predominantly assume unimodal signals, lacking cross-modal semantic consistency and robustness to missing modalities. Moreover, their evaluations are confined to simulated settings, typically with clean graphs and stationary distributions, without denoising or shift robustness, limiting external validity. See Appendix C.1 for details.

## 6 CONCLUSIONS

We propose a novel framework (ALMEA) that improves alignment accuracy and robustness through semantically calibrated modality imputation and active learning. ALMEA synthesizes embeddings for missing modalities, refining entity representations to address semantic inconsistencies. Integrated with active learning, ALMEA iteratively optimizes the selection of high-quality pairs for improved low-resource settings alignment. Analytical experiments demonstrate ALMEA's alignment accuracy and robustness in low-resource scenarios with resilience to semantic inconsistency across KG.

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

## A  IMPACT STATEMENT

We employed large language models (LLMs) as tools for grammatical refinement, polishing the precision of word choice and the appropriateness of content expression. LLMs enhanced grammatical accuracy, textual clarity, and lexical selection. No aspects of the research design, implementation process, data analysis, or conceptual development relied on LLMs. And, this work advances algorithmic methods for multimodal KG entity alignment and is evaluated in controlled settings on public MMKG benchmarks. Our objective is to improve accuracy and robustness; we do not collect new human-subject data or process personally identifiable information. Potential risks include unintended linkage/de-anonymization across datasets and the propagation of modality-specific or demographic biases. To mitigate these risks, we restrict experiments to licensed datasets with documented consent/usage terms, avoid sensitive attributes where possible, audit alignment errors for systematic bias, and release code/configurations (not pretrained models) with guidance discouraging

deployment on sensitive data without additional review. We report compute/energy budgets and prioritize efficient training/inference settings. While we do not foresee acute societal harms from the present study, we acknowledge these failure modes and outline safeguards to promote responsible use.

## B    FRAMEWORK DETAILS

### B.1    DERIVATION DETAILS

Tables 10–15 summarize the notations used throughout this paper, covering the multimodal knowledge graph definition, modal embedding construction, latent semantic modeling, active learning modules, and evaluation metrics.

### B.2    ADDITIONAL LOSSES DETAILS

For the completeness, the following illustrates the additional loss design of ALMEA: **Latent Distribution Normalization.**  The semantic normalization task aligns latent distributions with a Gaussian distribution to structure and smooth the latent space. To achieve this, we use the distribution matching loss (DML) Kingma (2013) to regulate the latent distribution:

$$\mathcal{L}_{DML} = \tfrac{1}{2} \sum_{j=1}^{d} \left( (\sigma^{m_j})^2 + (\mu^{m_j})^2 - \log((\sigma^{m_j})^2) - 1 \right) \qquad (9)$$

where $d$ is the dimension of the modality in the latent space, and $\mu^{m_j}$ and $\sigma^{m_j}$ represent the mean and standard deviation of the $j$-th dimension for modality $m$.

**Missing Modality Reconstruction.**  The LSL layer synthesizes the missing modality embedding within the original feature space, with reconstruction occurring after calibration. To achieve this, the missing latent variable $\tilde{z}^m$ of modality $m$ is generated using the reparameterisation trick and transformed into the latent embedding $\tilde{\mathbf{z}}^m$ via a calibration layer (Section 3.2). The reconstructed feature embedding $\hat{h}^m$ is then obtained via the decoder to approximate the original feature embedding $h^m$, completing the imputation for each modality $m$. The reconstruction loss (URL) for each modality $m$, measuring the discrepancy between the original $h^m$ and the reconstructed embeddings $\hat{h}^m$, is defined as follows:

$$\mathcal{L}_{URL} = \mathrm{MSE}(\mathrm{Decoder}(\tilde{\mathbf{z}}^m), h^m), \quad m \in \{g, r, a, v\}. \qquad (10)$$

The LSL layer learns to synthesize the missing modality embedding within the original feature space, with reconstruction occurring after the calibration layer (Section 3.2).

**Cross-Modal Alignment Loss.**    After semantic calibration, the VAE decoder reconstructs the embeddings of the modality $m$ features, which are fused via the cross-modal embedding aggregation function to produce joint embeddings $\hat{h}_S$ and $\hat{h}_T$ for the source and target MMKGs, respectively. The discrepancy between these embeddings is enforced by minimizing their discrepancy using Mean Squared Error as follows:

$$\mathcal{L}_{JAL}(\hat{h}_S, \hat{h}_T) = \mathrm{MSE}(\hat{h}_S, \hat{h}_T). \qquad (11)$$

where $\mathcal{L}_{JAL}$ minimizes discrepancies between the reconstructed joint embeddings to ensure semantic consistency across MMKGs.

### B.3    OPTIMIZATION FORMULATION AND DERIVATION

To learn efficiently from $Q_{\mathrm{sorted}}$ while minimizing selected candidate pairs within a limited budget, we formulate an optimization problem based on coding cost minimization. The coding cost quantifies subset representativeness, penalizing redundancy and maximizing value per selection. Let $\mathsf{z}_{(q_i)(q_j)} = 1$ indicate that $q_i$ represents $q_j$, with the encoding cost of $q_2$ using $q_i$ given by $a_{(q_i)(q_j)}$. With budget The optimization problem is formulated as follows:

$$\min_{Z_{ij}} \sum_{i=1}^{K} \lambda \|Z\|_{1,p} + \sum_{j=1}^{K} \sum_{i=1}^{K} a_{ij} \cdot Z_{ij} \quad \text{s.t.} \quad \sum_{i=1}^{K} Z_{ij} = 1, \quad \forall j; \quad Z_{ij} \in \{0, 1\}, \quad \forall i, j \qquad (12)$$

where the first term enforces sparsity on $Z$ via the $L_p$-norm, with $p$ controlling sparsification strength. The second term minimizes the total encoding cost over $Q_{\mathrm{sorted}}$. Constraints ensure each entity $e_j$ is assigned exactly one representative $e_i$, with $e_i = 0$ indicating $e_i$ cannot represent $e_j$.

**Relaxation.** Since the binary constraint renders the problem NP-hard, we relax it by allowing $\mathbf{z}_{ij} \in [0, 1]$ and $\mathbf{z}_{ij} \geq 0$, enabling a smoother optimization. Setting $p = 2$ with the $L_{2,1}$-norm further simplifies the convex optimization problem to:

$$\min_Z \lambda \|Z\|_{2,1} + \mathrm{tr}(A^T Z) \quad \text{s.t.} \quad 1^T Z = 1^T, \quad Z \geq 0 \tag{13}$$

where the first constraint ensures that each column of $Z$ sums to 1, indicating that each entity $e_j$ is represented by a weighted combination of other entities. The second constraint enforces non-negativity to ensure valid assignments.

**Augmented Lagrangian Formulation.** To solve this convex optimization problem, we use the Alternating Direction Method of Multipliers (ADMM) for iterative convergence. Specifically, an auxiliary variable $P$ is introduced to simplify the optimization problem in Eq.(8), which is then reformulated using the Augmented Lagrangian method as follows:

$$\min_{Z,P} \lambda \|DZ\|_{2,1} + \mathrm{tr}(A^T P) + \mu(Z - P) + \tfrac{\rho}{2}\|Z - P\|_F^2 \tag{14}$$

where $\mu \in \mathbb{R}^{d \times d}$ is the Lagrange multiplier for constraint enforcement, $\lambda$ controls row sparsity, and $\rho$ penalizes disagreement between $Z$ and the auxiliary matrix $P$ via $\|Z - P\|_F$. We optimize by alternating updates of $\{Z, P, \mu\}$. We calibrate sparsity via the Frobenius norm with the sparsification factor $\alpha$ as $\lambda = \frac{\sqrt{\sum_{i=1}^{K}\sum_{j=1}^{K} d_{ij}^2}}{\sqrt{K}} \cdot \alpha$, $\alpha \in [0.01, 1]$ where smaller $\alpha = 0.01$ yields milder sparsification while $\alpha = 1$ enforces the strongest row selection Boyd & Vandenberghe (2010).

**Derivation Details.** Applying the Augmented Lagrangian method, the problem is further expressed as:

$$\min_{Z,P} \lambda \|DZ\|_{2,1} + \mathrm{tr}(A^T P) + \mu(Z - P) + \frac{\rho}{2}\|Z - P\|_F^2 \tag{15}$$

Derivation of the last two terms in the optimization process for updating $Z$:

$$
\begin{aligned}
\mu(Z - P) + \frac{\rho}{2}\|Z - P\|_F^2 &= \mu(Z - P) + \frac{\rho}{2}\|Z - P\|_F^2 + \frac{1}{2\rho}\|\mu\|_F^2 - \frac{1}{2\rho}\|\mu\|_F^2 \\
&= \frac{\rho}{2}\left(\|Z - P\|_F^2 + 2\frac{\mu}{\rho}(Z - P) + \frac{\|\mu\|_F^2}{\rho^2}\right) - \frac{\|\mu\|_F^2}{2\rho} \\
&= \frac{\rho}{2}\left\|Z - P + \frac{\mu}{\rho}\right\|_F^2 - \frac{\|\mu\|_F^2}{2\rho}
\end{aligned} \tag{16}
$$

Derivation of the last three terms in the optimization process for updating $P$:

$$\text{tr}(A^T P) + \mu(Z - P) + \frac{\rho}{2}\|Z - P\|_F^2 = \text{tr}(A^T P) + \frac{\rho}{2}\|Z - (P - \frac{\mu}{\rho})\|_F^2$$

$$= \text{tr}(A^T P) + \frac{\rho}{2}\text{tr}[(Z - (P - \frac{\mu}{\rho}))^T(Z - (P - \frac{\mu}{\rho}))]$$

$$= \text{tr}(A^T P) + \frac{\rho}{2}\text{tr}[Z^T Z - 2Z^T(P - \frac{\mu}{\rho}) +$$

$$(P - \frac{\mu}{\rho})^T(P - \frac{\mu}{\rho})]$$

$$= \text{tr}(A^T P) - \text{tr}[\rho Z^T(P - \frac{\mu}{\rho})] +$$

$$\text{tr}[\frac{\rho}{2}((P - \frac{\mu}{\rho})^T(P - \frac{\mu}{\rho}))]$$

$$= \text{tr}(A^T P) - \rho\text{tr}(Z^T P) + \frac{\rho}{2}\text{tr}(P^T P) - \text{tr}(\mu^T P)$$

$$= \text{tr}[(A - \rho Z - \mu)^T P] + \frac{\rho}{2}\text{tr}\left(P^T P\right) \qquad (17)$$

$$= \frac{\rho}{2}\text{tr}[2\frac{1}{\rho}(A - \rho Z - \mu)P + P^T P]$$

$$= \frac{\rho}{2}\text{tr}(P^T P + 2\frac{1}{\rho}(A - \rho Z - \mu)P) +$$

$$tr[(\frac{1}{\rho}(A - \rho Z - \mu))^T(\frac{1}{\rho}(A - \rho Z - \mu))] -$$

$$tr[(\frac{1}{\rho}(A - \rho Z - \mu))^T(\frac{1}{\rho}(A - \rho Z - \mu))]$$

$$= \frac{\rho}{2}\|P + (\frac{1}{\rho}(A - \rho Z - \mu))\|_F^2$$

$$= \arg\min_P(\|P - (Z + \frac{\mu - A}{\rho})\|_F^2)$$

## B.4 FINAL UPDATE FORMULAS

**(Update of $Z$):**

$$Z^{(t+1)} = \arg\min_Z \lambda\|DZ\|_{2,1} + \mu^{(t)}(Z - P^{(t)}) + \frac{\rho}{2}\|Z - P^{(t)}\|_F^2$$

$$= \arg\min_Z \lambda\|DZ\|_{2,1} + \frac{\rho}{2}\|Z - P^{(t)} + \frac{\mu^{(t)}}{\rho}\|_F^2 - \frac{\|\mu^{(t)}\|_F^2}{2\rho} \qquad (18)$$

$$= \arg\min_Z(\frac{\lambda}{\rho}\|DZ\|_{2,1} + \frac{1}{2}\|Z - P^{(t)} + \frac{\mu^{(t)}}{\rho}\|_F^2)$$

where $Z^{(t+1)}$ is updated while $\{P, \mu\}$ are fixed.

In order to strengthen the convergence effect during the optimization process, we use $+A$ instead of $-A$ during the optimization process to prevent oversparsity and loss of critical information.

**(Update of $P$):**

$$P^{(t+1)} = \arg\min_P \text{tr}(A^T P) + \mu^{(t)}(Z^{(t+1)} - P) + \frac{\rho}{2}\|Z^{(t+1)} - P\|_F^2, \quad \text{s.t.} \quad 1^T P = 1^T, \quad P \geq 0$$

$$= \arg\min_P(\|P - (Z^{(t+1)} + \frac{\mu^{(t)} + A}{\rho})\|_F^2), \quad \text{s.t.} \quad 1^T P = 1^T, \quad P \geq 0$$

$$(19)$$

where $P^{(t+1)}$ is updated while $\{Z, \mu\}$ are fixed, with two constraints on $P$.

**(Update of $\mu$):**

$$\mu^{(t+1)} = \mu^{(t)} + \rho(Z^{(t+1)} - P^{(t+1)}) \qquad (20)$$

where $\mu^{(t+1)}$ is updated while $\{Z, P\}$ remain fixed.

### B.5 ALMEA ALGORITHM

The overall ALMEA framework is summarized in Algorithm 2, where the model $\mathbf{M}$ quantitatively learns from true entity pairs in the labelled set $L$ and updates the parameters. After completing the training phase, the model $\mathbf{M}$ iteratively selects representative and diverse candidate pairs from $Q$, adding them to $L$ based on their dissimilarity ranking oder in $Q_{sorted}$.

---

**Algorithm 2** ALMEA: Active Learning for MMEA

---

1: **Input:** $L = (\mathcal{G}_S^L, \mathcal{G}_T^L)$, $U = (\mathcal{G}_S^U, \mathcal{G}_T^U)$, number of total epochs $T_{total}$, base-training epochs $T_{base}$, ACS interval $T_{inter}$, number of ACS rounds $T_r$, max_iterations $T_{iter}$
2: **Output:** Updated labeled set $L$ **Constraint:** $T_{total} = T_{warm} + T_{inter} \times T_r$
3: Initialize model $\mathbf{M}$ (LSL & LSC) with $L, \mu, \rho, \alpha, \tau$ for pair selection
4: **Function** $ActiveCandidateSelection(\mathbf{M}, L, U)$:
5:     $Q_{\text{sorted}} \leftarrow$ Predict potential pairs from $\mathbf{M}$, Eq. 3
6:     $Z \leftarrow$ Initialize sparse matrix, Eq. 4
7:     $A \leftarrow$ Generate dissimilarity matrix, Eq. 5
8:     $C \leftarrow$ Calculate diversity, Eq. 6
9:     $D \leftarrow$ Generate diversity score, Eq. 7
10:    $P \leftarrow$ Initialize auxiliary matrix, Eq. 14
11:    **for** $t = 1$ to $T_{iter}$ **do**
12:       Update $Z^{(t+1)}$ using Eq. 18
13:       Update $P^{(t+1)}$ using Eq. 19
14:       Update $\mu^{(t+1)}$ using Eq. 20
15:       **if** $\|Z^{(t+1)} - P^{(t+1)}\|_F \leq \epsilon$ **then break**
16:    **end for**
17:    **return** $Z^*$
18: **for** $t = 1$ to $T_{total}$ **do**
19:    Train model $\mathbf{M}$ on $L$
20:    **if** $t \geq T_{warm} \wedge (t - T_{warm}) \bmod T_{interval} = 0 \wedge t < T_{total}$ **then**
21:       $q \in Q \cap ActiveCandidateSelection(\mathbf{M}, L, U)$
22:       $q_{\text{selected}} = \{q_i \mid q_i \geq \tau\}$
23:       $U \leftarrow U \setminus q_{\text{selected}}$
24:       $L \leftarrow L \cup q_{\text{selected}}$
25:    **end if**
26: **end for**

---

### B.6 TIME COMPLEXITY ANALYSIS

We analyze the computational cost of the active candidate selection (ACS) module. Computing distances between entities $h_{S_M} \in \mathcal{E}_S$ and $h_{S_N} \in \mathcal{E}_T$, requires $O(M \cdot N \cdot d)$, where $M = |\mathcal{E}_S|$, $N = |\mathcal{E}_T|$, and $d$ is the embedding dimension. For the unlabeled pools $\mathcal{E}_S^U = \{h_{S_1}, \ldots, h_{S_{M^U}}\}$ and $\mathcal{E}_T^U = \{h_{T_1}, \ldots, h_{T_{N^U}}\}$, the complexity is $O(M^U \cdot (M^U d + N^U d))$, yielding a total cost for Eq. 3 of $O(MNd + M^U(M^U d + N^U d)) = O((M^2 + 2(M^U)^2)d)$.

Constructing the dissimilarity matrix $A = \text{Avg}(A_S, A_T)$ in Eq. 5 requires $O(K^2 d)$, with $K = |Q_{\text{sorted}}|$. For labeled pools $\mathcal{E}_S^L$ and $\mathcal{E}_T^L$, computing the diversity matrix $C = \text{Avg}(C_S, C_T)$ has complexity $O(KM^L d)$. Diversity scores $D$ (Eq. 7) cost $O(KL)$.

The optimization problem in Eq. 14 decomposes into three updates: (i) $Z$-update by block soft-thresholding, (ii) $P$-update, and (iii) multiplier update, each $O(K^2)$. Thus, one iteration costs $O(K^2)$, and the overall optimization cost is $O(T_{\text{iter}} K^2)$. Combining all terms and noting $M = M^L + M^U$, the overall time complexity is

$$O\big(((M^L + M^U)^2 + 2(M^U)^2)d + K^2 d + KM^L d + T_{\text{iter}} K^2\big).$$

Hence, the dominant factors are the number of source entities $M$, candidate pairs $K$, and iterations $T_{\text{iter}}$. Complexity grows quadratically in $M$ and $K$. In practice, limiting $T_{\text{iter}}$ and pruning $Q_{\text{sorted}}$ (e.g.,

retaining only the top 50% of high-confidence pairs) substantially reduces computational overhead while maintaining accuracy.

Table 3: Dataset Summary. $|GT|$ represents the ground-truth entity alignments, indicating the number of aligned entity pairs between the source and target datasets. Specifically, $|GT|$(FB15K-DB15K) denotes the number of aligned entity pairs between FB15K and DB15K, while $|GT|$(FB15K-YAGO15K) denotes the aligned pairs between FB15K and YAGO15K. Some source entities may lack image modality or a corresponding target entity.

| $\mathcal{G}$ | $|\mathcal{E}|$ | $|\mathcal{R}|$ | #Triples($\mathcal{R}$) | $\mathcal{A}$ | #Triples($\mathcal{A}$) | $|\mathcal{V}|$ | $|GT|$(FB15K-DB15K) | $|GT|$(FB15K-YAGO15K) |
|---|---|---|---|---|---|---|---|---|
| FB15K | 14,951 | 1,345 | 592,213 | 116 | 29,385 | 13,444 | 12,846 | 11,199 |
| DB15K | 12,842 | 279 | 89,197 | 225 | 48,080 | 12,837 | 12,846 | – |
| YAGO15K | 15,404 | 32 | 122,886 | 7 | 23,532 | 11,194 | – | 11,199 |

## C  ADDITIONAL QUANTITATIVE STUDY

### C.1  ACTIVE LEARNING SETTING

The motivation for active learning arises from the prohibitive cost of large-scale annotation in real-world scenarios. While unlabeled data is typically abundant Sener & Savarese (2017), annotation requires substantial human effort, resources, and expense Mosqueira-Rey et al. (2023). Examples include rare-language speech recognition Settles (2009) and radiological diagnosis Yan et al. (2011).

The core challenge in MMEA is to align semantically related entities across heterogeneous MMKGs given limited contextual and multimodal evidence Li et al. (2023b). Performance degrades sharply under scarce supervision. To mitigate this, **ALMEA** embeds an active learning component that leverages unlabeled data to enrich semantics from a small annotated pool. Following standard protocols, we randomly label 20%, 50%, and 80% of the training set as the initial pool and treat the remainder as unlabeled, thereby simulating realistic annotation budgets. We refer to this setup as *simulated active learning*, and we explicitly contrast it with traditional active learning and data augmentation in Table 4.

### C.2  STATISTICAL SIGNIFICANCE TEST

To further evaluate the efficacy of **ALMEA**, we conducted paired t-tests Student (1908); Demšar (2006) against all benchmark models. As shown in Table 5, improvements achieved by ALMEA are statistically significant in most cases, rather than attributable to random fluctuations. MSNEA and OTMEA demonstrate separate features: they demonstrate significant fluctuations at 50% and 80% data splits. For instance, under the MRR, Hits@1, and Hits@10 metrics, MSNEA showed an average standard deviation of ±5.97%, OTMEA ±4.58%, whereas ALMEA maintained a low deviation of ±0.47%. Such pronounced fluctuations cannot be demonstrated to be statistically significant in experimental settings.

Moreover, ALMEA consistently demonstrated statistically significant superiority over most benchmark models in paired t-tests.

### C.3  SENSITIVITY ANALYSIS OF $\alpha$

We categorize experiments by data split ratio into low-resource (20(1) Under low-resource settings, ACS substantially improves alignment accuracy by prioritizing representative and diverse candidate pairs for annotation (Figures 5(a), 5(d)). In contrast, in high-resource settings (Figures 5(c), 5(f)), the labeled pool already covers sufficient information, leaving fewer high-confidence candidates in the unlabeled pool and thus limiting ACS's gains. This matches the intended role of ACS in alleviating low-resource challenges. (2) Across all six experimental groups (Figures 5(a–f)), different sparsification factors $\alpha$ were used: for FB15K-DB15K, $\alpha = \{0.35, 0.45, 0.95\}$ (a–c), and for FB15K-YAGO15K, $\alpha = \{0.25, 0.35, 0.45\}$. (d–f). As the resource ratio increases, larger $\alpha$ (stronger sparsification) is required, indicating that the model benefits from more aggressive redundancy filtering in richer regimes. These results further validate ACS's role in enhancing alignment robustness.

Table 4: Performance comparison among training strategies, including active learning, simulated active learning, and data augmentation.

| Paradigm | Active Learning | Simulated Active Learning | Data Augmentation |
|---|---|---|---|
| Data source | Large-scale, raw and uncurated unlabeled data Settles (2009). | Publicly available, cleaned large-scale labeled datasets Margatina & Aletras (2023). | Extended or synthesized versions of cleaned labeled datasets Shorten & Khoshgoftaar (2019). |
| Annotation cost | Annotation costs are prohibitively high, requiring oracles (human annotation, experts, etc.). | Manually hiding labels to simulate high annotation costs. | No additional annotation required. |
| Purpose | Through an iterative approach to selecting the most valuable samples for annotation, we maximize model performance while optimally utilizing annotation resources within limited budgets Hoi et al. (2006). | Similar to Active Learning, it simulates constrained experimental environments with limited budgets and maximizes model performance through iteration, enabling reproducible evaluation of active learning algorithms Margatina & Aletras (2023). | While maintaining label consistency, it enhances dataset diversity and model robustness solely by extending or synthesizing existing annotated data Cubuk et al. (2019). |
| Advantage | By iteratively selecting the most valuable samples within the tightest possible budget, the model minimizes access to costly labeling processes Settles (2009). The model adapts to noise and low-quality influences in new real-world settings (e.g., medical imaging and natural language processing), thereby enhancing its generalization capability and robustness Hoi et al. (2006). | The model ensures high reproducibility in controlled experimental settings, enabling fair comparisons with baseline models to validate robustness Margatina & Aletras (2023). Systematically shielding portions of the dataset through simulation evaluates sampling strategy effectiveness without incurring actual annotation costs Lowell et al. (2018). | Diverse samples are generated through transformations and modifications to address model overfitting Shorten & Khoshgoftaar (2019). Results are obtained using existing annotated data without requiring additional annotation or simulation Cubuk et al. (2019). |
| Limitation | Model performance gains may be inferior to random sampling Lowell et al. (2018). Policy effectiveness cannot be pre-validated, necessitating additional collection data for comparison—thereby negating the core objective of reducing annotation in active learning paradigms Attenberg & Provost (2011). | Strong model coupling means actively sampled datasets may fail to transfer to new datasets Tomanek & Morik (2011). Using high-quality datasets that have been cleaned and filtered significantly reduces noise and quality issues, potentially leading to an overestimation of simulated AL performance in real-world environments Margatina & Aletras (2023). | Often conducted in an ad-hoc manner, with explanations often remaining superficial at the level of regularization and lacking robust theoretical underpinnings Feng et al. (2021). Its benefits prove limited when applied to powerful pre-trained models or domain-specific tasks Shorten & Khoshgoftaar (2019). |

## C.4 Effectiveness of Active Learning

Apart from MRR, we also utilized True Positive Rate (TPR) and False Positive Rate (FPR) to measure the classification of positive and negative samples. Figures 9 show the ratio of True Positive (TP) and False Positive (FP) rates is analyzed based on temperature coefficients $\tau \in [0, 1]$ and truncation $q_{\text{selected}}$ across three sparsification factor $\alpha = \{0.1, 0.35, 1\}$, both without (a–c) and with (d–f) the diversity score $D$.

### C.4.1 Accuracy w/o Diversity Perspective.

As shown in Figures 9(a)-(c), TPR reaches its maximum value of nearly 36% at $\alpha = 0.1$ and $\tau = 0.75$, while at $\alpha = 1$, TPR peaks at approximately 41% for $\tau = 0.76$. This suggests that,

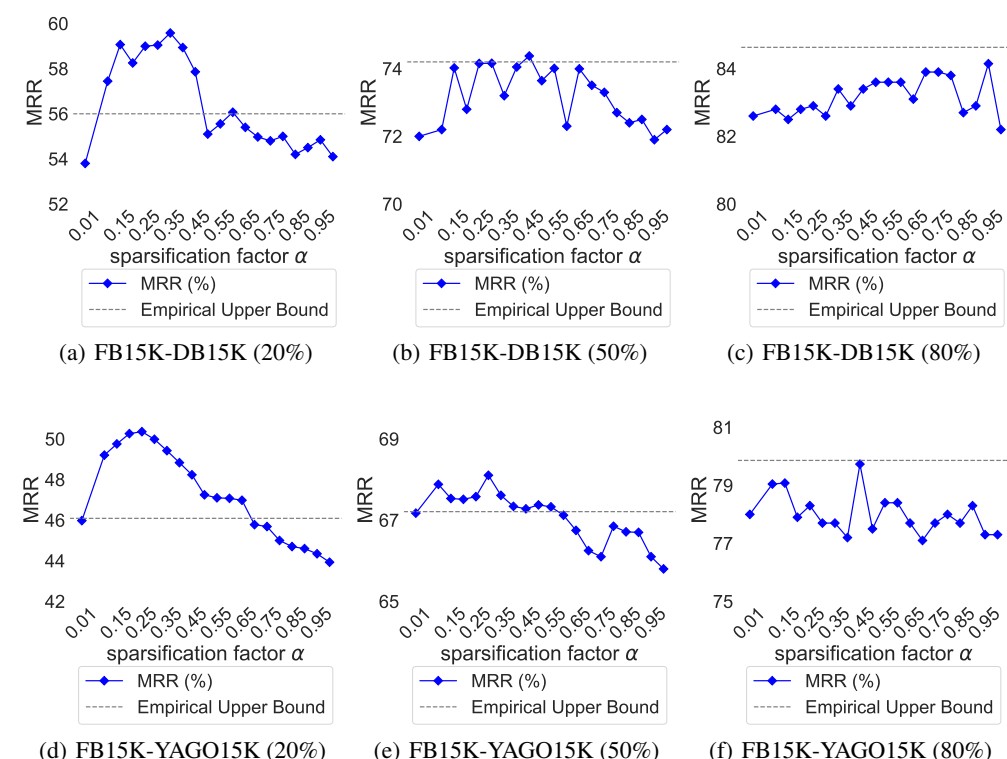

Figure 5: Sensitivity Analysis of $\alpha$. The ratio of MRR is analyzed based on sparsification factor $\alpha \in [0.01, 1]$. The empirical upper bound corresponds to the MRR of ALMEA w/o ACS, achieving 56.00%, 74.20%, and 84.63% on FB15K-DB15K (a-c) when the seed pairs are set to 20%, 50%, and 80%, respectively. And, for FB15K-YAGO15K (d–f), the MRR values are 46.09%, 67.21%, and 79.86%, respectively.

without the constraints imposed by $D$, the selected candidate pairs lack diversity, limiting the overall effectiveness of the selection process.

### C.4.2 ACCURACY WITH DIVERSITY PERSPECTIVE.

As shown in Figures 9(d)-(f), the TPR significantly improves, reaching maximum values of 91%, 99%, and 99% in $\alpha = \{0.1, 0.35, 1\}$ at $\tau = 0.01$. This indicates that the labeled potential pairs incorporate highly representative information as the sparsification direction is adjusted.

### C.5 EFFICIENCY ANALYSIS

In Table 6, we report the model parameter scales and training speeds. Compared to baseline models, **ALMEA w/o ACS** and **ALMEA** possess larger parameter counts, owing to the introduction of independent variational encoder encoders for each modality, thereby adding additional parameters. Furthermore, the ACS component incorporates an additional iterative process for screening candidate pairs, further increasing the parameter overhead. Nevertheless, the training speed of **ALMEA w/o ACS** and **ALMEA** remains superior to the vast majority of recent baseline models and state-of-the-art methods. This advantage stems primarily from the LSC component, which achieves completion of missing modal embeddings through simple MLP layers, while the Cross-Modal Calibration Loss effectively constrains and reconstructs these embeddings, thereby simplifying ALMEA's overall framework design.

# D  QUALITATIVE STUDIES

The dynamic weighting mechanism of MEAformer is susceptible to missing modalities, particularly visual and attribute, which leads its multi-head attention to amplify noisy cues from unrelated sources as proxies for absent semantic signals (e.g., "Zachary Quinto"). In FB15K, "Zachary Quinto" connects to 190 entities, while in DB15K, only 8 neighbors are present, introducing a sharp structural disparity and significant semantic inconsistency. Consequently, MEAformer tends to overemphasize the visual modality during alignment, failing to account for the semantic gaps induced by cross-KG heterogeneity. This imbalance renders it more prone to alignment errors, as it cannot adequately reweight modality contributions in the presence of structural and semantic divergence. Based on the case studies in Figure 6, we summarize the key limitations of existing MMEA approaches as follows:

- Figure 6(a): For the entity "Mayor", the source KG (FB15K) contains 190 unique neighbors but lacks the attribute modality, whereas the target KG (DB15K) has only 39 neighbors. This imbalance, combined with the missing modality, leads MEAformer to over-rely on the relational modality, misaligning the entity to 'Member Of Parliament'. In contrast, ALMEA reconstructs the missing attribute features and captures neighborhood-level inconsistencies, resulting in a globally balanced weighting and correct alignment to 'Mayor'.

- Figure 6(b): For the entity "River Thames", MEAformer fails due to conflicting neighborhood distributions and noisy visual modality features. It over-assigns high weights to attribute modalities, even though attribute modalities do not bring valid information, leading to incorrect alignment. ALMEA identifies and reduces the weights of inconsistent visual cues through semantic calibration, leading to accurate alignment.

- Figure 6(c): For "Perth Airport", despite minor visual discrepancies, MEAformer exhibits the same misalignment behavior as in previous cases: overfitting to the relational modality due to inconsistent neighborhood structures, leading to misclassification as Los Angeles International Airport. ALMEA effectively calibrates the semantic weights and utilizes consistent cross-KG clues to align correctly.

## D.1  QUALITY OF SYNTHESIZED MODALITY

To evaluate the quality of synthesized modalities, we use 20% of the training set and compare ALMEA with GEEA, which employs a VAE to align source and target knowledge graphs. Table 7 shows that both ALMEA and GEEA effectively reconstruct visual embeddings, but ALMEA outperforms GEEA in reconstructing neighbors and attributes. ALMEA's reconstruction for Perth Airport finds one more correct neighbor and two additional potential matches than GEEA. Additionally, ALMEA shows greater flexibility in reconstructing abstract semantics, such as "Australia."

Table 8 shows another example of the quality of calibrated embeddings for missing modalities in the compressed space after the LSL layer for the entity Bulgaria. As shown, ALMEA's attribute reconstruction also provides a broader range of potential correct matches than GEEA's reconstruction.

## D.2  MODALITY PREFERENCES

We compare instance-invariant global weights (ALMEA) with instance-conditional dynamic weights (MEAformer) for multimodal entity alignment. Using 20% of the training data, we analyze four hand-validated cases to probe ranking quality and modality attribution. As illustrated in Figure 8, dynamic weighting often collapses onto a single modality, yielding brittle rankings under missing/noisy signals. In the *River Thames* case, MEAformer places maximal mass on the attribute modality while under-weighting graph/relational cues, misranking the Mississippi and Columbia as the top candidates. By contrast, ALMEA's globally calibrated fusion maintains cross-modal balance, correctly ranking the Thames and the Rhine and producing sharper, semantically consistent attributions. These results suggest that global weights offer a stronger inductive bias in low-resource regimes and mitigate attention instability. In dynamic weighting, the Thames in the source knowledge graph has the highest proportion of attribute modalities (nine neighbors, six relationships, and four attributes), while the Thames in the target knowledge graph has a higher proportion of visual and graph modalities (12 one-hop neighbors, 10 relational modalities, and three attributes). This semantic inconsistency leads

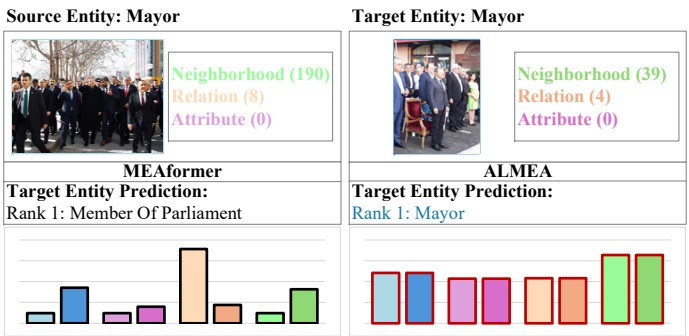

(a) Misweighting caused by missing attribute modalities and excessive neighbor disparities.

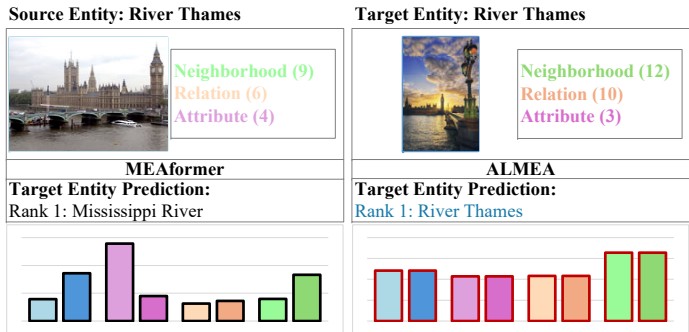

(b) Visual semantic inconsistencies cause over-reliance on irrelevant attribute modalities.

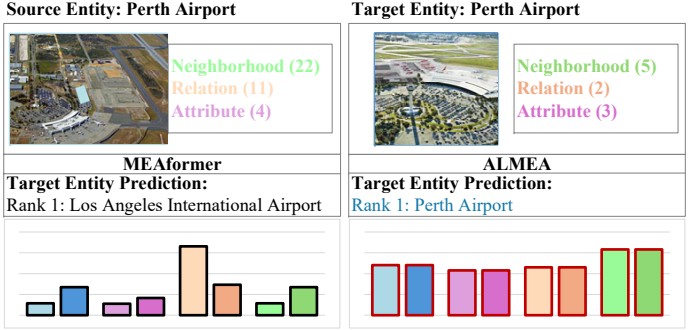

(c) Semantic inconsistencies in neighbor and relationship modalities suppress crucial visual information.

Figure 6: Motivating case studies compare MEAformer and ALMEA under varied modality-level semantic inconsistencies, highlighting challenges like missing modalities, and modality disparities.

to alignment failure in dynamic weighting. Our proposed global weighting approach effectively balances the semantic scale discrepancies, resulting in a successful alignment.

# E    ADDITIONAL ABLATION STUDY

## E.1    MASKING SENSITIVITY

**Setup.** To evaluate the robustness of our proposed model at different masking rates, we conducted a mask sensitivity analysis. The objective was to identify the optimal masking rate by minimizing standard deviation, ensuring a balance between performance and robustness.

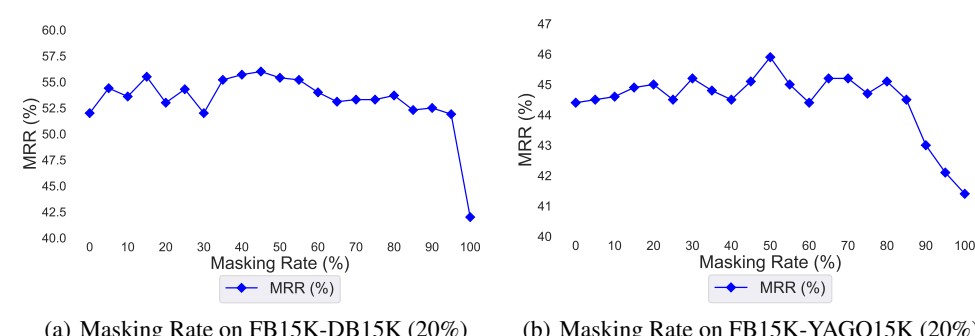

(a) Masking Rate on FB15K-DB15K (20%)  (b) Masking Rate on FB15K-YAGO15K (20%)

Figure 7: Analysis of Masking Rates on FB15K-DB15K and FB15K-YAGO15K datasets.

The masking rate varied from 0% to 100% in increment of 5%, resulting in 21 different settings. For each rate, the model was run five times to calculate the mean and standard deviation while keeping the other parameters fixed. Experiments were carried out on two datasets, FB15K-DB15K and FB15K-YAGO15K, using a training set 20%.

**Results.** The experimental results are presented in Figures 7(a) and 7(b). As the masking rate increases, the model performance (MRR, Hits@1, Hits@10) generally declines for both the FB15K-DB15K and FB15K-YAGO15K datasets, with a sharper performance drop and increased standard deviation, particularly beyond a masking rate 90%.

For FB15K-DB15K, the standard deviation of MRR, Hits@1, and Hits@10 is minimized at a masking rate 45%. While this is not the highest-performing setting, it offers a balanced trade-off between performance and robustness, with MRR, Hits@1, and Hits@10 reaching 56.00%, 46.81% and 73.52%, respectively. In contrast, for FB15K-YAGO15K, both the lowest standard deviation and the highest average performance occur at a 50% masking rate, yielding MRR, Hits@1, and Hits@10 values of 46.09%, 36.89%, and 63.81%, respectively.

# F    ADDITIONAL EXPERIMENTAL DETAILS

## F.1    BENCHMARK DATASET DETAILS

We evaluate the MMEA task on two widely recognized benchmark MMKGs: FB15K-DB15K and FB15K-YAGO15K Oñoro-Rubio et al. (2019); García-Durán & Niepert (2018); Liu et al. (2019). Each MMKG includes relational Liu et al. (2019), visual Oñoro-Rubio et al. (2019), textual, and numerical attributes García-Durán & Niepert (2018). Relations and Attributes represent unique relationship and attribute modality information within the complete dataset. Since an entity can have multiple attributes, attribute triples may appear multiple times, as do relationship triples. Ground-truth (GT) refers to the true entity alignments, where each source entity is uniquely paired with a target entity. Note that some source entities may lack image modality or a corresponding target entity. Entities lacking image information are pre-processed through random sampling using a normal distribution to ensure consistent input for the visual modality. The overall dataset statistics are summarized in Table 3.

## F.2    BASELINE DETAILS

For classical KG embedding methods, we consider the following approaches.

- **TransE**Bordes et al. (2013): Learns entity and relation embeddings in a continuous space, aligning entities through vector addition.

- **GCN-Align**Wang et al. (2018): Integrates structural and attribute information by mapping entities into a unified embedding space via a graph convolutional network.

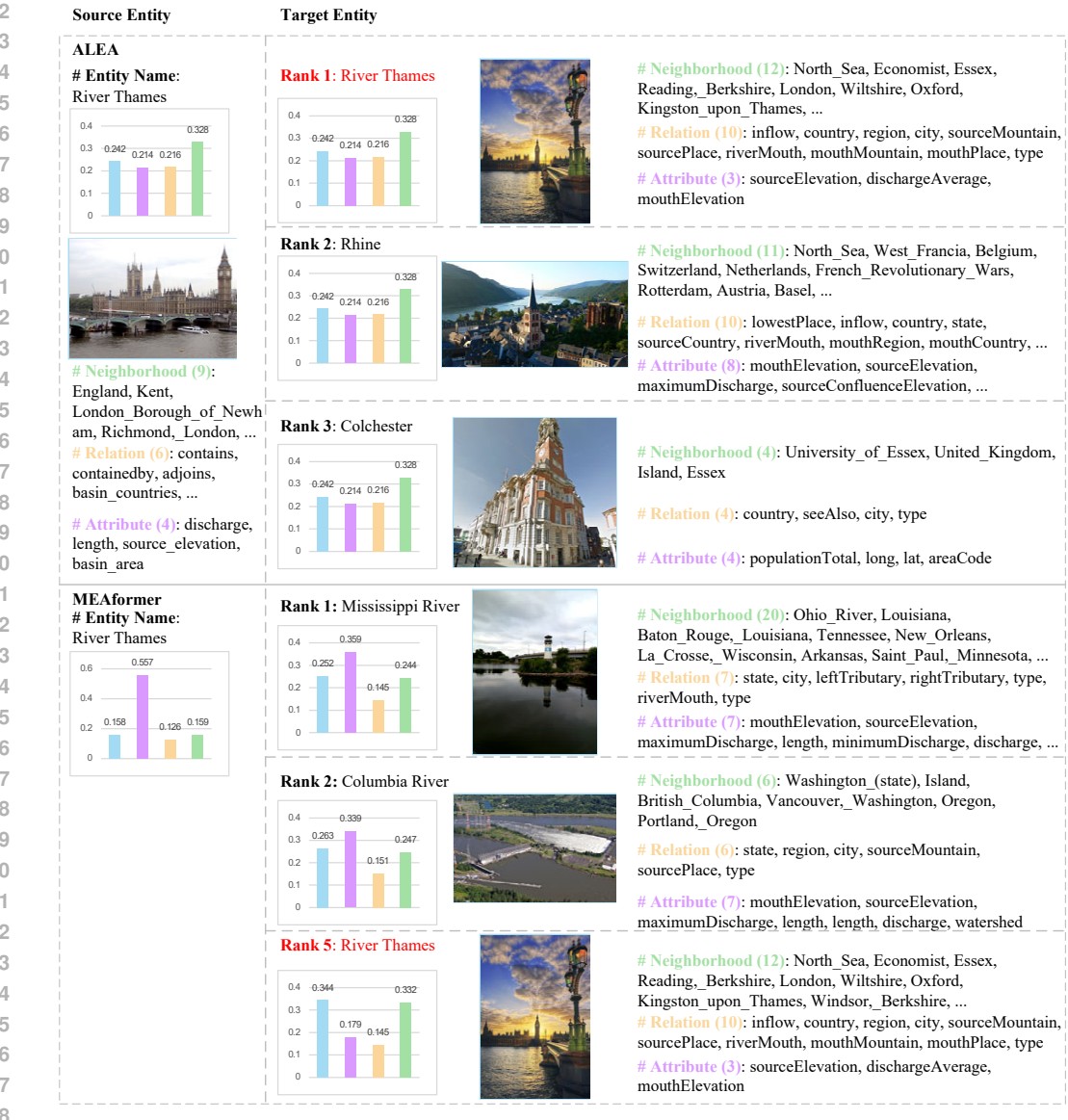

Figure 8: Qualitative Results on FB15K-DB15K. The upper part is our prposed method, and lower part is the SoTA MEAformer; the figure contains four modalities, namely visual, attribute, relational, and graph structure modalities, and the bar graph shows the weights of each modality of the joint modality. The red font represents the ranking of correct matches.

- **SEA**Pei et al. (2019): Enhances entity embeddings using degree-aware adjustments and optimizes alignment performance through iterative training with both labeled and unlabeled entity pairs.

For multimodal KG embedding methods, we consider the following approaches.

- **MMEA**Chen et al. (2020a): Maps visual, attribute, and relational modalities into a unified low-dimensional embedding space, and fuses these modalities to improve alignment accuracy.

- **EVA**Liu et al. (2021b): Emphasizes visual similarity as a key alignment factor, leveraging visual, relational, and attribute modalities for joint representation and unsupervised learning.

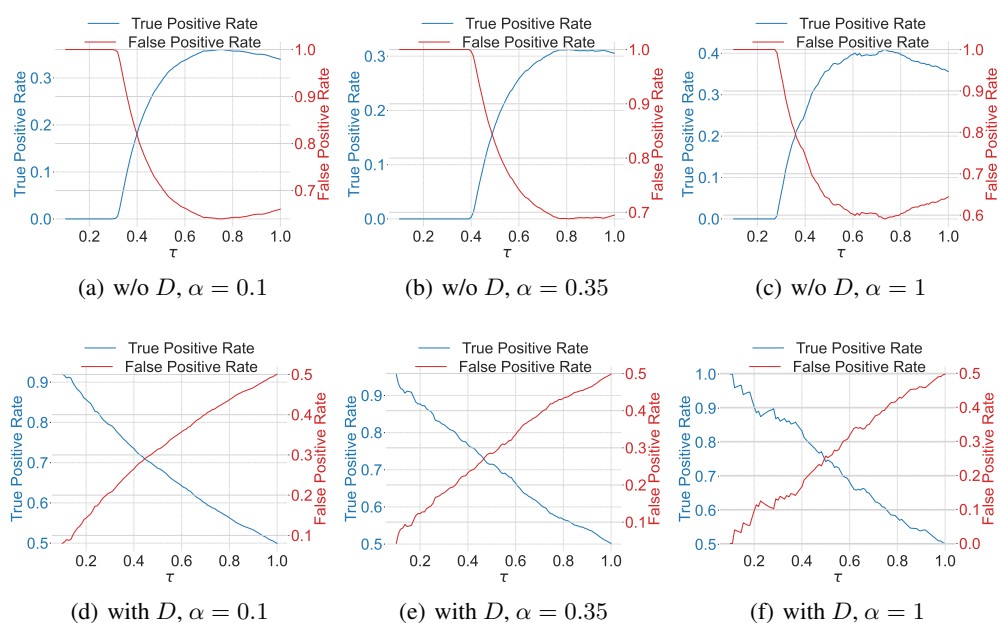

Figure 9: Effectiveness of Active Learning. The ratio of True Positive (TP) to False Positive (FP) rates is analyzed based on temperature coefficients $\tau \in [0, 1]$ and truncation $q_{\text{selected}}$ across three sparsification factor $\alpha = \{0.1, 0.35, 1\}$, both without (a–c) and with (d–f) the diversity score $D$.

- **MSNEA** Chen et al. (2022): Utilizes visual modalities to enhance relational and attribute-based alignment, focusing on cross-modal interactions.
- **MCLEA** Lin et al. (2022): Employs contrastive learning Chen et al. (2020b) to improve alignment by capturing intra-modal mutual information and cross-modal distribution differences, thereby strengthening modal fusion consistency.
- **GEEA** Guo et al. (2023): Uses VAE-based generative models to reconstruct embeddings across KGs, enhancing information sharing and feature associations to improve alignment.
- **MEAformer** Chen et al. (2023a): Introduces a multi-head attention mechanism Vaswani (2017) to dynamically adjust modality weights, mitigating information loss during fusion and extracting core semantics for optimized alignment.
- **OTMEA** Wang et al. (2025): Mitigates semantic heterogeneity between modalities through optimal transport (OT) Cao et al. (2022), specifically by mapping visual, neighbourhood, attribute, and relational modalities into a unified distributional space. This approach aims to approximate the geometric structure between modalities, thereby learning more robust joint embeddings.
- **SimDiff** Li et al. (2024): By mapping neighbourhood, attribute, relational, and visual modalities onto a unified latent space, perturbation modal embeddings are obtained through a Gaussian noise augmentation paradigm. Subsequently, a progressive denoising process implemented with stacked linear layers facilitates intermodal information transfer and thereby mitigates modal heterogeneity.

### F.3 IMPLEMENTATION DETAILS.

Training is conducted for $T_{total} = 750$ epochs, with an initial base-training phase of $T_{warm} = 500$ epochs followed by $T_r = 5$ active-learning rounds triggered every $T_{inter} = 50$ epochs ($T_{total} = T_{warm} + T_{inter} \times T_r$); as shown in Table 9(a). Experiments were conducted on a Tesla A100 GPU. The hyperparameters were optimized by grid search, yielding $\alpha = 0.35, \tau = 0.01, \rho = 0.1, \sigma = 3$, respectively as shown in Table 9(b). For models that could not be executed locally, including TransE, GCN-Align, SEA, and MMEA, we refer to the reported results from Chen et al. Chen et al. (2020a).

The results for EVA, MSNEA, MCLEA, GEEA, and MEAformer were obtained from our local experimental runs using default parameter settings. In our experiments, the loss coefficients ($\lambda_1$, $\lambda_2$, $\lambda_3$, and $\lambda_4$) are set to 1, 1, 1, and 1, respectively. The VAE module comprises a unimodal encoder, a hidden layer, and a unimodal decoder, each with dimensionality 300, and a joint modal decoder also of dimensionality 300. The learning rate is fixed at 0.001 with Adam optimization, a batch size of 3500, and a constant scheduler. ALMEA is implemented in PyTorch 2.2.2+cu121 with Python 3.12.1.

Table 5: Paired t-test between ALMEA variants and baselines over five runs. Each unit displays the $p$-value and t-statistic, with the upper portion representing the $p$-value and the lower portion representing the t-statistic. The significance threshold denotes statistical significance at different confidence levels, where $p < 0.05$ indicates significance, $p < 0.01$ denotes high significance, and $p < 0.001$ signifies extreme significance.

| Methods | FB15K-DB15K (20%) | | | FB15K-DB15K (50%) | | | FB15K-DB15K (80%) | | |
|---|---|---|---|---|---|---|---|---|---|
| | MRR | H@1 | H@10 | MRR | H@1 | H@10 | MRR | H@1 | H@10 |
| **ALMEA w/o ACS vs baselines** | | | | | | | | | |
| MCLEA | < 0.001 (145.09) | < 0.001 (139.21) | < 0.001 (81.08) | < 0.001 (29.23) | < 0.001 (30.80) | < 0.001 (9.73) | < 0.001 (40.04) | < 0.001 (50.92) | < 0.001 (9.96) |
| MSNEA | < 0.001 (157.38) | < 0.001 (231.82) | < 0.001 (82.06) | < 0.05 (4.06) | < 0.05 (3.77) | < 0.05 (4.01) | > 0.05 (2.12) | > 0.05 (2.24) | > 0.05 (1.93) |
| GEEA | < 0.001 (242.23) | < 0.001 (172.57) | < 0.001 (59.48) | < 0.001 (88.88) | < 0.001 (126.08) | < 0.001 (32.40) | < 0.001 (16.76) | < 0.001 (26.02) | < 0.01 (8.25) |
| MEAformer | < 0.001 (94.50) | < 0.001 (66.86) | < 0.001 (13.03) | < 0.001 (37.67) | < 0.001 (58.65) | < 0.001 (11.16) | < 0.001 (31.93) | < 0.001 (35.90) | < 0.05 (4.27) |
| OTMEA | < 0.01 (5.04) | < 0.01 (4.70) | < 0.01 (5.35) | < 0.01 (6.16) | < 0.01 (6.18) | < 0.01 (6.28) | < 0.05 (4.21) | < 0.05 (3.92) | < 0.01 (5.57) |
| SimDiff | < 0.001 (11.90) | < 0.001 (14.90) | < 0.05 (3.37) | < 0.001 (22.51) | < 0.001 (41.45) | < 0.05 (4.24) | < 0.001 (32.14) | < 0.001 (18.37) | < 0.05 (4.18) |
| **ALMEA vs baselines** | | | | | | | | | |
| MCLEA | < 0.001 (247.98) | < 0.001 (247.41) | < 0.001 (134.28) | < 0.001 (37.75) | < 0.001 (30.27) | < 0.001 (11.14) | < 0.001 (33.00) | < 0.001 (40.96) | < 0.01 (8.12) |
| MSNEA | < 0.001 (165.68) | < 0.001 (187.39) | < 0.001 (128.68) | < 0.05 (4.06) | < 0.05 (3.85) | < 0.05 (3.72) | > 0.05 (1.81) | > 0.05 (1.86) | > 0.05 (2.07) |
| GEEA | < 0.001 (238.55) | < 0.001 (358.78) | < 0.001 (88.81) | < 0.001 (96.13) | < 0.001 (118.02) | < 0.001 (25.62) | < 0.001 (13.11) | < 0.001 (22.34) | < 0.001 (6.88) |
| MEAformer | < 0.001 (129.86) | < 0.001 (131.04) | < 0.001 (29.52) | < 0.001 (34.66) | < 0.001 (39.33) | < 0.001 (17.07) | < 0.001 (21.27) | < 0.001 (21.19) | < 0.01 (4.92) |
| OTMEA | < 0.001 (6.63) | < 0.001 (6.76) | < 0.01 (6.21) | < 0.01 (6.32) | < 0.01 (6.01) | < 0.01 (6.89) | < 0.05 (3.76) | < 0.05 (3.40) | < 0.01 (5.92) |
| SimDiff | < 0.001 (39.66) | < 0.001 (45.74) | < 0.001 (20.65) | < 0.001 (21.23) | < 0.001 (29.08) | < 0.001 (5.46) | < 0.05 (2.34) | < 0.05 (3.56) | < 0.05 (1.79) |

| Methods | FB15K-YAGO15K (20%) | | | FB15K-YAGO15K (50%) | | | FB15K-YAGO15K (80%) | | |
|---|---|---|---|---|---|---|---|---|---|
| | MRR | H@1 | H@10 | MRR | H@1 | H@10 | MRR | H@1 | H@10 |
| **ALMEA w/o ACS vs baselines** | | | | | | | | | |
| MCLEA | < 0.001 (68.34) | < 0.001 (55.57) | < 0.001 (67.41) | < 0.001 (103.29) | < 0.001 (103.69) | < 0.001 (32.58) | < 0.001 (22.20) | < 0.001 (24.09) | < 0.001 (17.14) |
| MSNEA | < 0.001 (8.73) | < 0.001 (8.22) | < 0.001 (8.97) | < 0.01 (4.71) | < 0.01 (4.95) | < 0.05 (4.18) | < 0.05 (3.55) | < 0.05 (3.85) | < 0.05 (2.54) |
| GEEA | < 0.001 (179.46) | < 0.001 (117.18) | < 0.001 (134.81) | < 0.001 (235.04) | < 0.001 (165.13) | < 0.001 (57.81) | < 0.001 (36.57) | < 0.001 (36.78) | < 0.001 (31.20) |
| MEAformer | < 0.001 (45.51) | < 0.001 (36.81) | < 0.001 (44.11) | < 0.001 (66.52) | < 0.001 (103.69) | < 0.001 (17.76) | < 0.001 (20.24) | < 0.001 (28.11) | < 0.001 (13.91) |
| OTMEA | < 0.01 (6.61) | < 0.01 (5.07) | < 0.01 (7.11) | < 0.05 (3.99) | < 0.05 (3.42) | < 0.01 (5.31) | < 0.05 (2.20) | < 0.05 (2.58) | < 0.05 (2.60) |
| SimDiff | < 0.01 (4.84) | < 0.01 (3.16) | < 0.01 (6.46) | < 0.001 (57.95) | < 0.001 (64.89) | < 0.001 (17.31) | < 0.001 (33.26) | < 0.001 (45.40) | < 0.001 (33.19) |
| **ALMEA vs baselines** | | | | | | | | | |
| MCLEA | < 0.001 (178.12) | < 0.001 (120.99) | < 0.001 (181.67) | < 0.001 (235.65) | < 0.001 (149.31) | < 0.001 (20.83) | < 0.01 (7.73) | < 0.01 (6.70) | < 0.01 (7.31) |
| MSNEA | < 0.001 (14.21) | < 0.001 (13.72) | < 0.001 (13.97) | < 0.01 (5.49) | < 0.01 (5.85) | < 0.05 (4.43) | < 0.05 (2.85) | < 0.05 (3.06) | < 0.05 (2.11) |
| GEEA | < 0.001 (318.76) | < 0.001 (216.27) | < 0.001 (276.07) | < 0.001 (375.82) | < 0.001 (186.38) | < 0.001 (60.75) | < 0.001 (18.27) | < 0.001 (20.27) | < 0.001 (10.79) |
| MEAformer | < 0.001 (175.86) | < 0.001 (104.58) | < 0.001 (160.99) | < 0.001 (132.77) | < 0.001 (160.68) | < 0.001 (48.01) | < 0.01 (8.16) | < 0.01 (6.63) | < 0.01 (6.93) |
| OTMEA | < 0.001 (17.66) | < 0.001 (17.46) | < 0.001 (14.88) | < 0.01 (5.55) | < 0.01 (5.18) | < 0.01 (6.39) | > 0.05 (1.78) | > 0.05 (1.96) | > 0.05 (2.36) |
| SimDiff | < 0.001 (79.13) | < 0.001 (68.08) | < 0.001 (57.78) | < 0.001 (102.54) | < 0.001 (84.73) | < 0.001 (37.74) | < 0.01 (6.56) | < 0.05 (3.94) | < 0.01 (8.08) |

Table 6: Efficiency comparison. Parameter count and training speed on FB15K-DB15K (20% supervision). Higher Iter/s indicates faster training.

| Methods | Params (M) | Iter/s ↑ |
|---|---|---|
| MCLEA | 9.06 | 3.94 |
| GEEA | 47.46 | 0.73 |
| MEAformer | 10.58 | 1.01 |
| OTMEA | 11.02 | 0.69 |
| SimDiff | 10.31 | 0.68 |
| **ALMEA w/o ACS** | **14.34** | **1.20** |
| **ALMEA w ACS** | **14.34** | **1.05** |

Table 7: Qualitative Results of Entity Reconstruction on FB15K: Blue text highlights correctly reconstructed nearest neighbors, while underlined text indicates potential nearest neighbors.

| Ground Truth Nearest Neighbor | ALMEA Nearest Neighbor | GEEA Nearest Neighbor |
|---|---|---|
| **# Entity Name**: Perth Airport
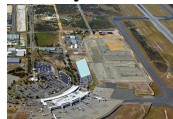 | **# Entity Name**: Perth Airport
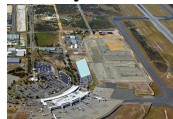 | **# Entity Name**: Perth Airport
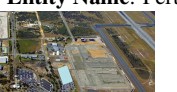 |
| **# Neighborhood**: Qantas, Jakarta, Cairns, Darwin_Northern_Territory, Canberra, Brisbane | **# Neighborhood**: Qantas, Jakarta, Cairns, Darwin_Northern_Territory, Canberra, Brisbane, Nine_Network, Network_Ten, Keith_Urban, Australia_(2008_film), Adelaide | **# Neighborhood**: Brisbane, Qantas, Cairns, Nine_Network, Network_Ten, University_of_Adelaide |
| **# Attribute**: number_of_runways, longitude, mean_elevation, latitude | **# Attribute**: number_of_runways, longitude, mean_elevation, latitude, wasCreatedOnDate | **# Attribute**: number_of_runways, longitude, mean_elevation, latitude, formationYear |

Table 8: Qualitative Results of Entity Reconstruction on FB15K: Blue text highlights correctly reconstructed nearest neighbors, while underlined text indicates potential nearest neighbors.

| Ground Truth Nearest Neighbor | ALMEA Nearest Neighbor | GEEA Nearest Neighbor |
|---|---|---|
| **# Entity Name**: Bulgaria
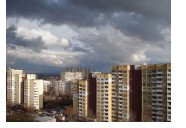 | **# Entity Name**: Bulgaria
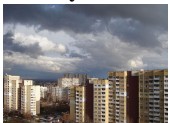 | **# Entity Name**: Bulgaria
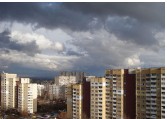 |
| **# Neighborhood**: Belgium, Hungary, Greece, Romania, Croatia | **# Neighborhood**: Hungary, Greece, Romania, Croatia, Slovenia, Chile, Colombia, Lithuania, Taiwan, Iceland | **# Neighborhood**: Belgium, Greece, Slovenia, Chile, Lithuania, Iceland |
| **# Attribute**: iso_numeric, longitude, latitude, area, date_founded, population_number, calling_code | **# Attribute**: iso_numeric, longitude, latitude, area, date_founded, population_number, calling_code, date_founded, date_dissolved, wasCreatedOnDate | **# Attribute**: iso_numeric, longitude, latitude, area, date_founded, population_number, calling_code, populationDensity |

Table 9: Hyperparameter settings in the main experiments. (a) Training Configuration Settings; (b) Loss Weights, LSC ang ACS hyperparameter Settings

**(a) Training Configuration Settings**

| Datasets | $T_{total}$ | $T_{base}$ | $T_{inter}$ | $T_r$ | $T_{iter}$ | batch-size | masked dropping rate | learning rate | sparsification factor $\alpha$ (20%, 50%, 80%) |
|---|---|---|---|---|---|---|---|---|---|
| FB15K-DB15K | 750 | 500 | 50 | 5 | 1000 | 3,500 | 0.45 | 0.001 | [0.35, 0.45, 0.95] |
| FB15K-YAGO15K | 550 | 300 | 50 | 5 | 1000 | 3,500 | 0.50 | 0.001 | [0.25, 0.35, 0.45] |

**(b) Loss Weights, LSC ang ACS hyperparameter Settings**

| Datasets | optimizer | loss weights $(\lambda_1, \lambda_2, \lambda_3, \lambda_4)$ | latent size (masked, unmasked) | activation function | MLP weight vector | $\rho$ | $\tau$ |
|---|---|---|---|---|---|---|---|
| FB15K-DB15K | Adam | 1,1,1,1 | 300,300 | Tanh | [300→300→1] | 0.1 | 0.01 |
| FB15K-YAGO15K | Adam | 1,1,1,1 | 300,300 | Tanh | [300→300→1] | 0.1 | 0.01 |

# G  NOTATION SUMMARY

To improve clarity and readability, we comprehensively summarize all mathematical notations used in this thesis. This includes notations related to multimodal knowledge graph structure, modal embedding construction, latent semantic learning and calibration, active candidate selection, and evaluation metrics.

Table 10: Multimodal Knowledge Graph (MMKG) Notations

| Notation | Explanation |
|---|---|
| $\mathcal{G}$ | Multimodal Knowledge Graph (MMKG) |
| $\mathcal{E}$ | Set of Entities in MMKG |
| $\mathcal{R}$ | Set of Relations in MMKG |
| $\mathcal{A}$ | Set of Attributes Associated with Entities |
| $\mathcal{V}$ | Set of Visual of Entities |
| $\mathcal{T}$ | Set of Relational Triples in MMKG |
| $\mathcal{G}_S, \mathcal{G}_T$ | Source and target Knowledge Graphs |

Table 11: Modal Embedding Construction (MEC) Notations

| Notation | Explanation |
|---|---|
| $x^g$ | Original Neighborhood Vectors |
| $x^r$ | Original Relation Vectors |
| $x^a$ | Original Attribute Vectors |
| $x^v$ | Original Visual Vectors |
| $h^g$ | Neighborhood Modality Embeddings |
| $h^r$ | Relation Modality Embeddings |
| $h^a$ | Attribute Modality Embeddings |
| $h^v$ | Visual Modality Embeddings |
| $W_g$ | Neighborhood Linear Transformation Weighting |
| $W_r$ | Relation Linear Transformation Weighting |
| $W_a$ | Attribute Linear Transformation Weighting |
| $W_v$ | Visual Linear Transformation Weighting |
| $\mathcal{M}$ | Pre-trained Multimodal Encoder |
| GAT | Graph Attention Network (GAT) Transformation |
| Concat | Concatenation Operation |

Table 12: Latent Semantic Learning (LSL) Notations

| Notation | Explanation |
| --- | --- |
| $\tilde{h}^g$ | Masked Neighborhood Modality Embeddings |
| $\tilde{h}^r$ | Masked Relation Modality Embeddings |
| $\tilde{h}^a$ | Masked Attribute Modality Embeddings |
| $\tilde{h}^v$ | Masked Visual Modality Embeddings |
| $\mathbf{h}^g$ | Hidden Outputs of Neighborhood Modality |
| $\mathbf{h}^r$ | Hidden Outputs of Relation Modality |
| $\mathbf{h}^a$ | Hidden Outputs of Attribute Modality |
| $\mathbf{h}^v$ | Hidden Outputs of Visual Modality |
| $z^g$ | Latent Representation of Neighborhood Modality |
| $z^r$ | Latent Representation of Relation Modality |
| $z^a$ | Latent Representation of Attribute Modality |
| $z^v$ | Latent Representation of Visual Modality |
| $\mathbf{z}^g$ | Latent Embeddings of Neighborhood Modality |
| $\mathbf{z}^r$ | Latent Embeddings of Relation Modality |
| $\mathbf{z}^a$ | Latent Embeddings of Attribute Modality |
| $\mathbf{z}^v$ | Latent Embeddings of Visual Modality |
| $\mathcal{N}(\mu, \sigma^2)$ | Gaussian Distribution with Mean $\mu$ and Standard Deviation $\sigma$ |
| $\epsilon \sim \mathcal{N}(0, 1)$ | Normal Distribution |
| $\mathcal{L}_{DML}$ | Distribution Matching Loss |
| $\mathcal{L}_{URL}$ | Unimodal Reconstruction Loss |
| Eecoder | Variational Autoencoder Encoder |
| Decoder | Variational Autoencoder Decoder |

Table 13: Latent Semantic Calibration (LSC) Notations

| Notation | Explanation |
| --- | --- |
| Tanh | Tanh Activation Function |
| MLP | Multi-Layer Perceptron |
| MSE | Mean Squared Error |
| $\mathcal{L}_{SCL}$ | Semantic Calibration Loss |
| $\mathcal{L}_{JAL}$ | Joint Reconstruction Loss |
| $\lambda_1, \lambda_2, \lambda_3, \lambda_4$ | Loss Weighting Factor |

Table 14: Active Candidate Selection (ACS) Notations

| Notation | Explanation |
| --- | --- |
| $Q$ | Candidate Pool of Potential Pairs |
| $Q_{\text{sorted}}$ | Sorted Candidate Pool Based on Confidence |
| $Z \in \mathbb{R}^{n \times n}$ | Sparse Matrix |
| $A \in \mathbb{R}^{n \times n}$ | Dissimilarity Matrix |
| $D \in \mathbb{R}^{n \times 1}$ | Diversity Score |
| $C \in \mathbb{R}^{n \times n}$ | Representativeness Score Matrix |
| $\alpha$ | Sparsification Factor |
| $\tau$ | Confidence Threshold |
| $P$ | Auxiliary Variable Matrix in ADMM Optimization |
| $k$ | Max Iteration for ADMM Optimization |
| $\mu$ | Lagrange Multiplier for ADMM Constraints |
| $\rho$ | ADMM Penalty Parameter |
| $\|\cdot\|_{2,1}$ | $L_{2,1}$-norm for Sparsification |
| $\|\cdot\|_F^2$ | The square of the Frobenius Norm |
| $\text{tr}(\cdot)$ | Trace Term |

Table 15: Evaluation Metrics Notations

| Notation | Explanation |
|----------|-------------|
| MRR | Mean Reciprocal Rank |
| Hits@1 | Hits at Rank 1 |
| Hits@10 | Hits at Rank 10 |
| TPR | True Positive Rate |
| FPR | False Positive Rate |

