# OpenReview forum: "ALMEA: Active Learning-Enhanced Multimodal Entity Alignment with Semantic Modality Imputation"
_ICLR.cc/2026/Conference — Submitted to ICLR 2026_

### Official Review · Reviewer_pQ3f · 2025-10-28

**Soundness:** 2
**Presentation:** 1
**Contribution:** 2
**Rating:** 4
**Confidence:** 4

**Summary:**

This paper proposes ALMEA, a novel framework for Multimodal Entity Alignment under low-resource settings. ALMEA integrates semantic imputation of missing modalities with active learning to improve alignment robustness. The authors evaluate ALMEA on two datasets (FB15K-DB15K and FB15K-YAGO15K) under varying supervision budgets, and the results show consistent improvements over state-of-the-art baselines.

**Strengths:**

This paper addresses two critical challenges in MMEA, missing modalities and limited seed alignments, both of which are common in real-world KGs but often overlooked in prior work.

The experiments cover multiple settings, including ablation studies, statistical significance tests, and qualitative case analyses.

The modular structure (LSL, LSC, ACS) enhances interpretability and facilitates component-wise analysis.

**Weaknesses:**

Leveraging VAE to generate the missing modalities has already been explored by GEEA and UMAEA [1]. The latter also proposes a new multi-modal OpenEA benchmark. The authors currently evaluate ALMEA only on two single-lingual datasets. It would be better to validate the effectiveness of the proposed method across multiple benchmarks.

This paper uses active learning for annotation, which is analogous to the bootstrapping or iterative algorithms used in existing EA and MMEA methods. However, the authors only present the non-iterative results of baselines. The performance advantage of the proposed method isn't that large.

The baselines (e.g., MEAFormer) are marked with "*" indicating reproduction. However, their results are significantly lower than those in their original paper. This is perhaps because the authors use a constant embedding dimension setting for all methods, but why?

The writing and presentation also need improvement to meet the standard of ICLR. For example, Figure 2 introduces too many terms that are not discussed (as well as Algorithm 1), and tables contain several typos.

[1] Rethinking Uncertainly Missing and Ambiguous Visual Modality in Multi-Modal Entity Alignment, ISWC 2023.

**Questions:**

Please see Weaknesses.

---

> ### Author Response · Authors · 2025-11-25
>
> **W1**
>
> Due to space limits, our anonymous link: [Multilingual MMKG](https://github.com/RTX4090123/ALMEA/blob/main/Multilingual%20MMKG%20Experiments.md)
>
> Although both ALMEA and GEEA employ VAEs, ALMEA introduces two key advances:
>
> 1. **Latent Space Calibration**
>
> ALMEA improves the expressiveness of the latent space through LSC, inspired by the ALBEF framework. This calibration procedure primarily addresses semantic inconsistencies, explicitly calibrate modal weights, as detailed in the new Figure 1(c). Unlike GEEA and UMAEA, which relies solely on the VAE decoder to reconstruct missing modalities, As a result, it achieves superior performance in bridging semantic gaps.
>
> 2. **Active Learning (ACS) Module**
>
> ALMEA introduces AL paradigm, which allows to select the most informative pairs.  Building on this, we further discover that with just 5\% budget, ACS can outperform ALMEA without ACS with higher training set by identifying the most informative pairs, achieving lower training costs. For instance, a 15\% training set combined with a 5\% budget yields superior results compared to a 20\% training set alone, as detailed in the new Figure 1(d)
>
> **Multilingual MMKG Experiments**
>
> Beyond Cross-KG Dataset, we have constructthe new DBP15K Dataset (ZH–EN, JA–EN, FR–EN).
>
> To ensure a comprehensive evaluation, we employed three settings:
>
> 1. **w/ SF (with surface form)**
>
> Under this setting, the dataset incorporates attribute and feature augmentation, which substantially enriches the model's expressive power. However, as demonstrated by many prior studies, this may introduce data leakage risks.
>
> 2. **w/o SF (without surface form)**
>
> To mitigate data leakage concerns, we employ the standard MMEA paradigm, evaluating ALMEA's alignment prediction capability across four modalities without semantic augmentation.
>
> 3. **Unsupervised setting**
>
> Following the EVA paradigm, we employ visual signals [2] to guide the model in matching entities based on image similarity, denoted as ALMEA-V. Following the MCLEA name-signal paradigm[1], we guide the model in matching entities based on name similarity, denoted as ALMEA-N.
>
> **Findings:**
>
> Our model outperforms all baselines across three settings. Due to time constraints, not all baselines fully reproduced; therefore, we follow SimDiff and report its reproduced scores for fair comparison.
>
> Experimental results shows that  the ACS component demonstrably improves performance across all three settings, achieving an average increase of 9.97% in the Hits@1 metric, because all three datasets being trained on 30% of the training set (low-resource scenario). Given that DBP15K's multimodal semantics are richer and of higher quality than those of cross-KGs, this directly enhances ACS's effectiveness on DBP15K. ALMEA operates effectively not only on English-only datasets but also in multilingual scenarios, further highlighting its robustness.
>
> **W2**
>
> Due to space limits, our anonymous link: [U B test](https://github.com/RTX4090123/ALMEA/blob/main/Unlimited%20Budget%20test.md)
>
> Through exploration, we discovered that Iterative Learning[1] and active learning appear mechanistically similar, as both employ bidirectional nearest neighbors for Bidirectional nearest neighbours (BNN) (Eq. (3)). In the current MMKG literature, we observe that SimDiff (SoTA), MEAformer, MCLEA, and MSNEA all employ the Iterative Learning Paradigm. When we reconstructed the original MEAformer paper, iterative learning was not enabled, which explains the significant performance divergence.
>
> **IL does not involve querying labels, the IL paradigm detail is as follows:**
> - BNN are computed once per small-cycle.
> - Within a big-cycle, multiple rounds of filtering are performed, retaining only pairs present in all filtering.
> - At the end of the big-cycle, all pairs are added to the training set as pseudo-labels.
> Thus, IL operates cleanly as self-training, unconstrained by annotation budgets and offering no guarantee of pseudo-label quality. Furthermore, the computational cost is relatively high due to the requirement to perform nearest neighbour embedding calculations at each iteration.
>
> **The ACS paradigm:**
> - BNN are computed only once per ACS iteration cycle.
> - Each ACS iteration scores candidate pairs based on representativeness and diversity settings.
> - Every big-cycle, only a specific number of the highest-value candidates pairs are labelled and added to the training set.
> - A clearly controllable annotation budget.
>
> Baseline models using the IL all operate under unlimited budget constraints. To ensure fairness and comparability under equal conditions, we have reclassified all IL-based baseline models to the unlimited budget setting. Within the unlimited budget settings, ALMEA outperforms all IL-based baselines, whilst ACS significantly enhances efficiency whilst circumventing the error propagation issues inherent in IL.
>
> **W4**
> We appreciate the attention, we have fixed the issue.
>
> [1] Sun et al., “BootEA”, IJCAI 2018

---

### Official Review · Reviewer_iYi6 · 2025-10-29

**Soundness:** 2
**Presentation:** 2
**Contribution:** 2
**Rating:** 4
**Confidence:** 4

**Summary:**

This paper proposes a novel framework called ALMEA for MMEA, which addresses the problem of aligning entities across MMKGs with missing modalities and semantic inconsistencies. The proposed approach integrates semantic calibration and active learning to improve alignment in low-resource scenarios, especially where manually annotated seed pairs are scarce. The ALMEA framework includes three core modules: Latent Semantic Learning, Latent Semantic Calibration, and Active Candidate Selection. Experimental results show that ALMEA outperforms existing baseline methods, demonstrating an improvement in alignment accuracy in various low-resource scenarios.

**Strengths:**

1. LSL effectively synthesizes embeddings for missing modalities, which ensures that the missing modality information can still contribute to alignment tasks.
2. The use of active learning to select the most informative entity pairs for annotation is a key strength, helping to alleviate the low-resource challenge by iteratively improving the alignment model with minimal manual intervention.
3. The paper provides strong experimental results across two benchmark datasets. ALMEA consistently outperforms the state-of-the-art baselines, especially in low-resource settings.

**Weaknesses:**

1. About novelty. One concern is that active learning has been used for weakly supervised EA, which is your mentioned low-resource scenarios, for a long time. It seems ALMEA is similar to these active learning-based methods [1,2]. Another concern is that some methods have similar performance to ALMEA in the same settings, like DESAlign [3], which is also designed for tackling the semantic consistency. So why don`t you compare and analyze them? Or maybe there are some other different settings I have ignored.
2. While the paper provides strong qualitative results, there is limited statistical validation to support the claim that ALMEA’s improvements are significant and not due to random variation. Error bars or significance tests like paired t-tests could provide more confidence in the reported improvements.
3. While the paper claims that global weighting is more robust, a more detailed analysis of how these different weighting strategies perform across various datasets would be beneficial. A more detailed comparison of the global weighting vs. dynamic weighting could further clarify the advantages of ALMEA's approach in different settings.


[1].Berrendorf, Max, Evgeniy Faerman, and Volker Tresp. "Active learning for entity alignment." European Conference on Information Retrieval. Cham: Springer International Publishing, 2021.

[2].Liu, Bing, et al. "ActiveEA: Active Learning for Neural Entity Alignment." 2021 Conference on Empirical Methods in Natural Language Processing, EMNLP 2021. Association for Computational Linguistics (ACL), 2021.

[3].Wang, Yuanyi, et al. "Towards semantic consistency: Dirichlet energy driven robust multi-modal entity alignment." 2024 IEEE 40th International Conference on Data Engineering (ICDE). IEEE, 2024.

**Questions:**

Please see the weakness.

---

> ### Author Response · Authors · 2025-11-25
>
> **W1**
>
> Due to space limits, our anonymous link: [U B test](https://github.com/RTX4090123/ALMEA/blob/main/Unlimited%20Budget%20test.md)
>
> [1] is one earliest works to introduce the Active Learning framework within the EA domain, aiming to select individual entities and prompt the annotator to identify all matching nodes in the graph. Fundamentally, the AL paradigm within EA operates solely on structural embedding spaces, confronting challenges primarily stemming from issues such as insufficient node edges and sparse structural embeddings.
>
> **Key Distinctions**
>
> **Active Learning in EA**
> 1. Scores and annotates only a single entity
> 2. Scoring relies solely on structural modalities
> 3. Low-resource settings in EA refer to sparse structural embeddings caused by insufficient node edges. For example, [2] applies margin-based uncertainty for node scoring but remains confined to node-level structural embeddings.
>
> **Active Learning in MMEA**
> 1. Operates on entity pairs without manual annotation — ACS automatically labels the most valuable candidate pairs.
> 2. Scoring is based on joint modalities within a multimodal shared embedding space.
> 3. Low-resource setting refers to challenges posed by missing modalities and noisy modalities, which exacerbate cross-modal semantic inconsistencies (weighting system imbalance). ALMEA ingeniously employs LSC to calibrate cross-modal semantic shifts, then leverages ACS to capture candidate pairs.
>
> **New Baseline DESAlign**
>
> By exploring DESAlign, it belongs to the iterative learning paradigm, not the AL paradigm. We discovered that iterative learning [4] shares a similar mechanism with active learning, both employing the bidirectional nearest neighbour algorithm (BNN) (Eq. (3)). Within the current MMKG literature, we observe that DESAlign, MEAformer, MCLEA, and MSNEA all adopt the iterative learning paradigm.
>
> **Key Distinctions**
>
> **IL does not involve querying labels, the IL paradigm detail is as follows:**
> - BNN are computed once per small cycle.
> - Within a big-cycle, multiple rounds of filtering are performed, retaining only pairs present in all filtering.
> - At the end of the big-cycle, all pairs are added to the training set as pseudo-labels.
> Thus, IL operates cleanly as self-training, unconstrained by annotation budgets and offering no guarantee of pseudo-label quality. Furthermore, the computational cost is relatively high due to the requirement to perform nearest neighbour embedding calculations at each iteration.
>
> **The ACS paradigm:**
> - BNN are computed only once per ACS iteration cycle.
> - Each ACS iteration scores candidate pairs based on representativeness and diversity settings.
> - Every big-cycle, only a specific number of the highest-value candidates pairs are labelled and added to the training set.
> - A clearly controllable annotation budget.
>
> All Baseline models using the IL all operate under unlimited budget constraints, including DESAlign. New baseline reproduction, Non-iterative for DESAlign also. To ensure fairness and comparability under equal conditions, we have reclassified all IL-based baseline models to the unlimited budget setting. Within the unlimited budget settings, ALMEA outperforms all IL-based baselines, whilst ACS significantly enhances efficiency whilst circumventing the error propagation issues inherent in IL. Non-iterative test remain inferior to ALMEA w/o ACS in DESAlign.
>
>
> **W2**
>
> **Summary of statistical significance (paired t-test over 5 runs)**
>
> | Dataset & Ratio     | MCLEA | MSNEA | GEEA | MEAformer | OTMEA | SimDiff(SoTA) |
> |---------------------|-------|-------|------|-----------|-------|---------------|
> | FB15K-DB15K (20%)   | ***   | ***   | **   | ***       | **    | **  |
> | FB15K-DB15K (50%)   | ***   | **    | ***  | ***       | **    | *** |
> | FB15K-DB15K (80%)   | ***   | *     | *    | ***       | —     | ***  |
> | FB15K-YAGO15K (20%) | ***   | ***   | ***  | ***       | **    | *** |
> | FB15K-YAGO15K (50%) | ***   | **    | ***  | ***       | —     | ***  |
> | FB15K-YAGO15K (80%) | ***   | *     | ***  | ***       | —     | *** |
>
>
> We have introduced the t-test in Table 5 of the appendix. This **Summary of statistical significance (paired t-test over 5 runs)** represents an optimized version, which we shall add into the main body. **Significance levels: \* = p < 0.05, \*\* = p < 0.01, \*\*\* = p < 0.001, — = not significant**
>
> **Key findings**
>
> 1. Across all datasets and training settings, ALMEA achieved t-test significance at p < 0.01 in the majority of tests, showing that alignment performance reflects statistical reliability rather than random variation.
>
> 2. p > 0.05 results only occurred with OTMEA, due to its markedly higher variation alignment accuracy compared to other benchmarks.
>
> **W3**
>
> The term “global weighting” should be referred to as “LSC weighting”. We have revised Figure 1 to illustrate the improvements brought by LSC and ACS more clearly.
>
> [4] Sun et al., “BootEA”, IJCAI 2018

---

### Official Review · Reviewer_jwT2 · 2025-10-30

**Soundness:** 3
**Presentation:** 3
**Contribution:** 3
**Rating:** 6
**Confidence:** 4

**Summary:**

This paper proposes a new MMEA framework called ALMEA to employ active learning for maintaining semantic consistency across different KGs and addressing the low-resource scenario. ALMEA consists of three designs: Latent Semantic Learning (LSL), atent Semantic Calibration (LSC), and Active Candidate Selection (ACS). Experiments are conducted on FB15K-DB15K and FB15K-YAGO15K.

**Strengths:**

- The active learning framework is firstly introduced in the MMEA field. Therefore, the design of the overall framework is novel.
- Low-resource scenario is an important topic for MMEA research.

**Weaknesses:**

- The citation format in this paper is wrong, which should be revised in the rebuttal. \cite --> \citep
- The paper shows that ALMEA has lower performance gain when the data is sufficient and it mainly works for low-resource scenario.
- The unsupervised MMEA setting is not explored in the experiments. Besides, the datasets used in the experiments are mainly FB15K-DB15K and FB15K-YAGO15K, which are monolingual (English) datasets. The multilingual datasets are not explored in the main experiments.
- The presemtation of Table 1 can be further optimized to make it more clear.

**Questions:**

- What about ALMEA's performance under unsupervised MMEA? Can active learning still work? I hope that you can add more experiments on this setting to show whether active learning works for it.
- In the datasets you used, the entity amounts are not significantly different. Can you consider other scenario that the entity amounts of the two KGs are with a significant order-of-magnitude gap?

---

> ### Author Response · Authors · 2025-11-25
>
> **W1**
>
> We thank the reviewers for their attention and have addressed all citation issues.
>
> **W4**
>
> We have revised Table 1.
>
> **Q1**
>
> Due to space limits, our anonymous link: [Multilingual MMKG](https://github.com/RTX4090123/ALMEA/blob/main/Multilingual%20MMKG%20Experiments.md)
>
> **Multilingual MMKG Experiments**
>
> Beyond FB15K-DB15K and FB15K-YAGO15K, we have restructured the code and hyperparameters to enable the model to be perfectly adapted to the new DBP15K benchmarks (ZH–EN, JA–EN, FR–EN).
>
> To ensure a comprehensive evaluation, we employed three settings:
>
> 1. **w/ SF (with surface form)**
>
> Under this setting, the dataset incorporates attribute and feature augmentation, which substantially enriches the model's expressive power. However, as demonstrated by numerous prior studies, this may introduce data leakage risks.
>
> 2. **w/o SF (without surface form)**
>
> To mitigate data leakage concerns, we employ the standard MMEA paradigm, evaluating ALMEA's alignment prediction capability across four modalities without semantic augmentation.
>
> 3. **Unsupervised setting**
>
> Following the EVA paradigm, we employ visual signals [2] to guide the model in matching entities based on image similarity, denoted as ALMEA-V. Following the MCLEA name-signal paradigm[1], we guide the model in matching entities based on name similarity, denoted as ALMEA-N.
>
> **Findings:**
>
> Our model outperforms all baselines across three settings. Due to time constraints, not all baselines fully reproduced; therefore, we follow SimDiff and report its reproduced scores for fair comparison.
>
> Experimental results shows that  the ACS component demonstrably improves performance across all three settings, achieving an average increase of 9.97% in the Hits@1 metric, because all three datasets being trained on 30% of the training set (low-resource scenario). Given that DBP15K's multimodal semantics are richer and of higher quality than those of cross-KGs, this directly enhances ACS's effectiveness on DBP15K. ALMEA operates effectively not only on English-only datasets but also in multilingual scenarios, further highlighting its robustness.
>
> **W2**
>
> 1. When training data is sufficiently abundant, the model has already fully learned multimodal semantics (entity nodes provide exceptionally rich semantic information), so the benefits of the active learning paradigm naturally diminish.
>
> 2. Resource-constrained scenarios are crucial in practical settings due to limitations in manual annotation. Consequently, model capable of tolerating missing entity modalities while autonomously identifying candidate pairs is essential. As shown in Figure 1(d), we randomly selected 20% and 30% of the training set as baselines. ACS paradigm configurations: 15% training data + 5% budget versus 25% training data + 5% budget. This validates the necessity of active learning paradigms in low-resource scenarios, a finding further corroborated by the new experimental results in Q1. The gradual diminishing returns observed in high-resource scenarios represent an inevitable outcome.
>
>
> **Q2**
>
> Due to space limits, our anonymous link: [Scale Imbalanced KG](https://github.com/RTX4090123/ALMEA/blob/main/Scale%20Imbalanced%20KG%20Setting.md)
>
> **Experimental Setup**
>
> For the FB15K–DB15K dataset, the total number of entities on the Source set is 14,951, while the Target set contains 12,842 entities. To simulate the disparity in scale between Source and Target, we reduced the total number of Target entities by factors of 0.5, 0.1, and 0.05, respectively, relative to the total on the left side. These are denoted as ALMEA (Imbalance-0.5×), ALMEA (Imbalance-0.1×), and ALMEA (Imbalance-0.05×), respectively.
>
> **Experimental Results**
>
> We observe minimal performance degradation when data imbalance is set to 0.5×. After randomly masking half of the Target entities, the ACS algorithm still finds it easier to predict matching entities from the Source KG to the Target KG. Most ground truth entities remain discoverable within the candidate space, as ACS filters out noise while preserving multimodal semantic information.
>
> When the Target total was reduced to 0.1×, most Source KG entities could no longer find matching counterparts in the Target KG. Constructing the dissimilarity matrix D became divergent and ambiguous, resulting in a significant decline in alignment performance. The same applies at 0.05×.
>
> [1] Lin, Z., Zhang, Z., Wang, M., Shi, Y., Wu, X., & Zheng, Y. (2022). Multi-modal contrastive representation learning for entity alignment.
> [2] Liu, F., Chen, M., Roth, D., & Collier, N. (2021, May). Visual pivoting for (unsupervised) entity alignment.

---

### Official Review · Reviewer_kymN · 2025-11-02

**Soundness:** 3
**Presentation:** 3
**Contribution:** 3
**Rating:** 6
**Confidence:** 3

**Summary:**

This paper proposes ALMEA, a novel framework for Multimodal Entity Alignment (MMEA) that combines semantic imputation and active learning to improve robustness under missing-modality and low-resource conditions. The method consists of three main modules:
(1) Latent Semantic Learning (LSL) uses a VAE-based generative model to synthesize embeddings for missing modalities;
(2) Latent Semantic Calibration (LSC) aligns cross-graph semantic distributions via KL divergence to mitigate semantic inconsistency;
(3) Active Candidate Selection (ACS) employs a diversity-regularized subset selection strategy to efficiently choose representative entity pairs under a limited labeling budget.
Experiments on FB15K–DB15K and FB15K–YAGO15K show consistent improvements over state-of-the-art baselines (up to +5.16% MRR and +5.57% Hits@1), especially in low-resource scenarios.

**Strengths:**

1. Novel integration of semantic imputation and active learning for MMEA, addressing both missing modalities and sparse supervision.

2. Comprehensive evaluation on two benchmark MMKG datasets with multiple baselines, demonstrating consistent gains in both low- and high-resource settings. And Clear modular design (LSL, LSC, ACS) that enables ablation and interpretability.

3. Strong robustness analysis, showing resilience to missing modalities and diverse candidate selection strategies.

**Weaknesses:**

1. The connection between the active learning optimization and overall alignment objective lacks formal analysis or proof of convergence.

2. Training involves multiple components (VAE, calibration, optimization with ADMM), which may increase computational cost and make real-world deployment difficult.

2. While quantitative results are strong, more case studies or visualization of latent semantics could better illustrate the improvements brought by LSC and ACS.

**Questions:**

1. Could the active learning module be combined with uncertainty-based criteria (e.g., entropy or margin sampling) to further improve sample efficiency?

2. What is the computational overhead compared to MEAformer or SimDiff, and how does ALMEA scale with large-scale MMKGs?

---

> ### Author Response · Authors · 2025-11-25
>
> **Q1**
>
> Due to space limits, our anonymous link: [ACS vs Uncertainty](https://github.com/RTX4090123/ALMEA/blob/main/ACS_vs_Uncertainty.md)
>
> As described in [1], active learning methods can be broadly categorized into uncertainty-based, representativeness-based, and diversity-based paradigms, with ACS representing a fusion of the latter two. To investigate whether such uncertainty signals benefit MMEA, we modified ACS as follows:
>
> **Margin Uncertainty-based Method:** Based on Eq.  (3), the core objective of ACS is to obtain $Q_{sorted}$. Thus, we define $d_i(j) = \|h_{S,i} - h_{T,j}\|^2$, where $j \in \{1,\ldots,N\}$ enumerates all target entities and $i \in \{1,\ldots,M\}$ enumerates all source entities. And then let the nearest and second-nearest targets be: $j_1 = \arg\min_j d_i(j), \quad j_2 = \arg\min_{j\ne j_1} d_i(j)$. So, we have distance-based margin is $Q_m(i)= d_i(j_2) - d_i(j_1)$. A smaller margin indicates higher alignment uncertainty, so we select candidates with the smallest margins[1].
>
> **Pseudo-entropy Uncertainty-based Method:** Since no probability distribution was used during the construction of Q, we adopt the entropy of the binomial Bernoulli distribution [2] to establish a pseudo-entropy uncertainty-based paradigm. We start from the distance, compute the nearest target distance as: $d_i^{\min}=\min_j d_i(j)$. We then normalize to the interval $[0,1]$ to obtain similarity scores: $\hat d_i = \mathrm{Norm}(d_i^{\min}) \in [0,1], \quad s_i = 1-\hat d_i$, where $s_i$ serves as pseudo-confidence for entity $i$, and we clamp $s_i$ into $[\varepsilon, 1-\varepsilon]$ for numerical stability. Finally, we define a Bernoulli-style pseudo-entropy for entity $i$: $Q_e(i) = -\big( s_i \log s_i + (1-s_i)\log(1-s_i) \big)$. So, we have a larger $Q_{entropy}(i)$, which signifies more hesitant and uncertain alignment decision.
>
> **Findings:**
>
> 1. **Margin-based uncertainty performs unexpectedly.** The algorithm selects the most uncertain candidate pairs, such as $(i, j)$—where the model cannot determine whether its prediction for $i$ corresponds to $j_1$ or $j_2$. Therefore, expert annotations are required to resolve the model's prediction ambiguity. This approach results in suboptimal performance of $Q_m$.
> 2. **The entropy-based outperforms the margin-based but remains low compared to the ACS method.** For the MMEA task, the entropy paradigm selects samples where the model is most hesitant, rather than those it finds most difficult to distinguish and least confident (as chosen by the margin-based method). Therefore, the entropy method significantly outperforms the margin-based approach. Qualitative analysis indicates that when selecting candidate pairs in a single-round ACS, the ACS method achieves a True Positive Rate (TPR) of 99.22%. In contrast, the entropy method yields a TPR of 95.34%. Although the difference is modest, we observe that the entropy method's selection from the 'hesitation region' inevitably introduces noise, further confirming its slightly inferior performance relative to the ACS method.
>
> **Q2**
>
> Due to space limits, our anonymous link: [Computational Cost](https://github.com/RTX4090123/ALMEA/blob/main/computational_cost.md)
>
> We have covered the computational cost comparison in Table 6, where we observe that our model possesses more parameters than other baseline models. For instance, 0.4 million more parameters than the SoTA model, yet achieves higher operational efficiency, running 0.76 iter/s faster than the SoTA.
>
> The overall ACS complexity is: $O(M^2d + (M^U)^2 + K^2 d + KM^Ld + KM^L + T_{\text{iter}} K^2)$, where $M$ denotes the total number of entities, $M^L$ and $M^U$ denote labelled and unlabelled subsets respectively, $K$ is the number of candidate pairs (with $K \ll M$), and $T_{iter}$ is the number of ADMM iterations.
>
> Both $K$ and $T_{iter}$ are controlledable hyperparameters and remain small in practice. Therefore, they can be safely ignored in asymptotic analysis. Combining all terms and noting that $M = M^L + M^U$ implies $M^L, M^U \le M$, we can upper-bound $(M^U)^2$ $M^L d$ and $M^L$ by $O(M^2 d)$, so we have $O\big(M^2d + Md+ M + M^2\big)$. Thus, the worst-case complexity is dominated by the $O(M^2 d)$ term, a quadratic term in $M$ rather than an exponential term, and is consistent with the Table 5.
>
> **W3**
>
> We have revised Figure 1 to illustrate the improvements brought by LSC and ACS more clearly.
>
> [1] Settles, B. (2009). Active learning literature survey.
>
> [2] Cover, T. M., & Thomas, J. A. (2006). Elements of Information Theory. Wiley.

---

### Meta-Review · Area_Chair_udic · 2026-01-10

**Summary:**

This paper receives the scores of 6,6,4,4 (average 5.0). The main concerns raised by the reviewers include:
- novelty overlap with prior VAE-based imputation (GEEA/UMAEA) and active learning in EA/MMEA,
- limited benchmarks (initially monolingual, no unsupervised) and questions on baseline reproduction;
- missing formal analysis/convergence for active learning and limited statistical significance in the main text;
- presentation issues and potential computational complexity for multi-component training.

**Reviewer Concerns:**

The authors add multilingual DBP15K experiments (w/ and w/o surface forms) and unsupervised variants (visual/name signals), provide paired t-tests across multiple runs, analyze ACS against uncertainty-based criteria, clarify computational costs/complexity, and distinguish ACS from iterative learning baselines (e.g., DESAlign).

These responses substantially strengthen empirical coverage and clarity, though formal convergence analysis and some novelty concerns remain only partially addressed.

**Reviewer Scores:**

Based on the reviewers' comments, the two reviewers who gave positive scores are likely to maintain their evaluations, while the two reviewers who gave negative scores are also likely to keep their scores unchanged, given that their concerns have not been fully addressed.

---

### Decision · Program_Chairs · 2026-01-26

Reject